

# Reviews and syntheses: Turning the challenges of partitioning ecosystem evaporation and transpiration into opportunities

Paul C. Stoy[1], Tarek El-Madany[2], Joshua B. Fisher[3,4], Pierre Gentine[5], Tobias Gerken[1,6], Stephen P. Good[7], Shuguang Liu[8], Diego G. Miralles[9], Oscar Perez-Priego[2,10], Todd H. Skaggs[11], Georg Wohlfahrt[12], Ray G. Anderson[11], Martin Jung[2], Wouter H. Maes[9], Ivan Mammarella[13], Matthias Mauder[14], Mirco Migliavacca[2], Jacob A. Nelson[2], Rafael Poyatos[15,16], Markus Reichstein[2], Russell L. Scott[17], Sebastian Wolf[18]

[1]Department of Land Resources and Environmental Sciences, Montana State University, Bozeman, MT, USA
[2]Max Planck Institute for Biogeochemistry, Hans Knöll Straße 10, Jena D-07745, Germany
[3]Jet Propulsion Laboratory, California Institute of Technology, 4800 Oak Grove Drive, Pasadena, CA 91109, USA
[4]Joint Institute for Regional Earth System Science and Engineering, University of California at Los Angeles, Los Angeles, CA 90095, USA
[5]Department of Earth and Environmental Engineering, Columbia University, New York, 10027, USA
[6]The Pennsylvania State University, Department of Meteorology and Atmospheric Science, 503 Walker Building, University Park, PA
[7] Department of Biological & Ecological Engineering, Oregon State University, Corvallis, Oregon, USA
[8]National Engineering Laboratory for Applied Technology of Forestry and Ecology in South China, Central South University of Forestry and Technology, Changsha, China
[9]Laboratory of Hydrology and Water Management, Ghent University, Coupure Links 653, 9000 Gent, Belgium
[10]Department of Biological Sciences, Macquarie University, North Ryde, NSW 2109, Australia
[11]U.S. Salinity Laboratory, USDA-ARS, Riverside, CA, USA
[12] Institut für Ökologie, Universität Innsbruck, Sternwartestr. 15, 6020 Innsbruck, Austria
[13]Institute for Atmospheric and Earth System Research / Physics, Faculty of Science, FI-00014 University of Helsinki, Finland
[14]Karlsruhe Institute of Technology, Institute of Meteorology and Climate Research - Atmospheric Environmental Research, Garmisch-Partenkirchen, Germany
[15]CREAF, E08193 Bellaterra (Cerdanyola del Vallès), Catalonia, Spain
[16]Laboratory of Plant Ecology, Faculty of Bioscience Engineering, Ghent University, Coupure links 653, 9000 Ghent, Belgium
[17]Southwest Watershed Research Center, USDA Agricultural Research Service, Tucson, AZ, USA
[18]Department of Environmental Systems Science, ETH Zurich, Zurich, Switzerland

*Correspondence to*: Tarek El-Madany (telmad@bgc-jena.mpg)

**Abstract.** Evaporation ($E$) and transpiration ($T$) respond differently to ongoing changes in climate, atmospheric composition, and land use. Our ability to partition evapotranspiration ($ET$) into $E$ and $T$ is limited at the ecosystem scale, which renders the validation of satellite data and land surface models incomplete. Here, we review current progress in partitioning $E$ and $T$, and provide a prospectus for how to improve theory and observations going forward. Recent advancements in analytical techniques provide additional opportunities for partitioning $E$ and $T$ at the ecosystem scale, but their assumptions have yet to be fully tested. Many approaches to partition $E$ and $T$ rely on the notion that plant canopy conductance and ecosystem water use efficiency (EWUE) exhibit optimal responses to atmospheric vapor pressure deficit ($D$). We use observations from 240 eddy covariance flux towers to demonstrate that optimal ecosystem response to $D$ is a reasonable assumption, in agreement with recent studies, but the conditions under which this assumption holds require further analysis. Another critical assumption for many $ET$ partitioning approaches is that $ET$ can be approximated as $T$ during ideal transpiring conditions, which has been



challenged by observational studies. We demonstrate that $T$ frequently exceeds 95% of $ET$ from some ecosystems, but other ecosystems do not appear to reach this value, which suggests that this assumption is ecosystem-dependent with implications for partitioning. It is important to further improve approaches for partitioning $E$ and $T$, yet few multi-method comparisons have been undertaken to date. Advances in our understanding of carbon-water coupling at the stomatal, leaf, and canopy level open

new perspectives on how to quantify $T$ via its strong coupling with photosynthesis. Photosynthesis can be constrained at the ecosystem and global scales with emerging data sources including solar-induced fluorescence, carbonyl sulfide flux measurements, thermography, and more. Such comparisons would improve our mechanistic understanding of ecosystem water flux and provide the observations necessary to validate remote sensing algorithms and land surface models to understand the changing global water cycle.

## 1 Introduction

Some 70 thousand cubic kilometers of water leave terrestrial ecosystems and enter the atmosphere through evapotranspiration ($ET$) every year (Jung et al., 2018; Oki and Kanae, 2006). Despite its importance, we are unsure whether global $ET$ has been increasing over time (Brutsaert, 2013, 2017; Brutsaert and Parlange, 1998; Zeng et al., 2018; Zhang et al., 2016) such that the water cycle is accelerating (Ohmura and Wild, 2002), or if it has been decreasing and causing more river discharge (Gedney

et al., 2006; Labat et al., 2004; Probst and Tardy, 1987). Reanalyses, upscaled estimates, and land surface model (LSM) outputs disagree with regards to global ET volumes (Mueller et al., 2013) by up to 50% (Mao et al., 2015; Vinukollu et al., 2011). LSMs also struggle to simulate the magnitude and/or seasonality of $ET$ at the ecosystem scale (Figure 1), suggesting fundamental gaps in our understanding of the terrestrial water cycle. These issues need to be resolved to effectively manage water resources as climate continues to change (Dolman et al., 2014; Fisher et al., 2017).

Along with technological and data limitations, we argue that a fundamental challenge in modeling $ET$ at the global scale is difficulty measuring soil and plant-intercepted evaporation ($E$) and transpiration ($T$) at the ecosystem scale (Fisher et al., 2017; McCabe et al., 2017). LSMs and remote sensing algorithms (see Appendix A) rely on process-based understanding of $E$ and $T$ to estimate $ET$, but it is not clear how to guide their improvement without accurate ground-based $E$ and $T$ observations at spatial scales on the order of a few kilometers or less (Talsma et al., 2018) and temporal scales that capture diurnal, seasonal,

and interannual variability. Interest in partitioning $E$ and $T$ from ecosystem $ET$ measurements has grown in recent years (Anderson et al., 2017b), and many new measurement and modeling approaches seek to do so, but often rely on assumptions that are not grounded in evidence. We begin with a brief research review that notes recent updates to our theoretical understanding of $ET$ and outlines the challenges in measuring $E$ and $T$ at the ecosystem scale. We then describe current and emerging innovations in partitioning $E$ and $T$ (Table 1) and challenge some of the assumptions that these approaches rely on

against observations. We finish with an outlook of how carefully-designed ecosystem-scale experiments can constrain models of E and T to improve our understanding going forward.



## 2 Background

### 2.1 Vegetation plays a central role in evaporation and transpiration partitioning

The ratio of transpiration to evapotranspiration ($T/ET$) is related to aridity (Good et al., 2017), but appears to be relatively insensitive to annual precipitation ($P$) (Schlesinger and Jasechko, 2014). $T/ET$ is sensitive to ecosystem characteristics, namely

the leaf area index ($LAI$) (Berkelhammer et al., 2016; Fatichi and Pappas, 2017; Wang et al., 2014), especially on sub-annual time scales (Li et al., 2019; Scott and Biederman, 2017), noting that LAI is related to $P$ at longer time scales. More LAI favors $T$ and $E$ from intercepted water ($E_i$) at the expense of soil $E$ ($E_{soil}$) such that $LAI$ explains some 43% of the variability of annual $T/ET$ across global ecosystems (Wang et al., 2014). Upscaling this relationship results in a global estimate of terrestrial annual $T/ET$ of $0.57 \pm 0.07$ (Wei et al., 2017). Other observational studies suggest that annual $T/ET$ averages nearly 2/3 globally (0.61

$\pm$ 0.15 (Schlesinger and Jasechko, 2014), $0.64 \pm 0.13$ (Good et al., 2015), and $0.66 \pm 0.13$ across some FLUXNET sites (Li et al., 2019)). Inter-comparison studies agree on the large uncertainty surrounding these estimates, with reported global terrestrial annual $T/ET$ ratios ranging from 0.35 to 0.90 (Coenders-Gerrits et al., 2014; Fatichi and Pappas, 2017; Young-Robertson et al., 2018). Approaches that use stable isotopes tend to produce higher annual T/ET values due to assumptions regarding isotopic fractionation (Jasechko et al., 2013; Sutanto et al., 2014). Some LSM estimates of annual $T/ET$ arrive at larger values on the

order of 0.70 $\pm$0.09 (Fatichi and Pappas, 2017; Paschalis et al., 2018), while other LSMs suggest smaller $T/ET$; for example, $T/ET$ from the IPCC CMIP5 intercomparison ranges from 0.22 to 0.58 (Wei et al., 2017). Constraining these model results with observations results in an estimate similar to observatoinsl studies but with reduced uncertainty: $0.62 \pm 0.06$ (Lian et al., 2018). An ongoing challenge is to measure and model $T/ET$ correctly at the ecosystem scale across all time scales over which it varies from minutes or less to multiple years or more. For this, an understanding of ecosystem water transport and biological

responses to micrometeorological forcing is necessary (Badgley et al., 2015).

### 2.2 Turning theory into practice

Measuring and modeling water fluxes from surface to atmosphere at the ecosystem scale across multiple scales in time is a non-trivial challenge. The pools in which water is stored in ecosystems span multiple scales from soil pores to forest canopies. Liquid and gaseous water transport occurs through pathways in the soil, xylem, leaves, and plant surfaces that exhibit nonlinear

responses to hydroclimatic forcing, which is itself stochastic (Katul et al., 2007, 2012). These complex dynamics of water storage and transport impact the conductance of water between ecosystems and atmosphere (Mencuccini et al., 2019; Siqueira et al., 2008), and these conductance terms are central to the Penman-Monteith equation, which combines the thermodynamic, aerodynamic, environmental, and biological variables to which $ET$ responds to represent the mass and energy balance of water flux between the land surface and the atmosphere (Monteith, 1965; Penman, 1948)

$$ET = \frac{1}{\lambda} \frac{s(R_n - G) + \rho_a c_p D g_a}{s + \gamma \left(1 + \frac{g_a}{g_{surf}}\right)}. \tag{1}$$

In the Penman-Monteith equation, $\lambda$ is the latent heat of vaporization (J g$^{-1}$), $s$ is the slope of the saturation vapor pressure function (Pa K$^{-1}$), $R_n$ is the surface net radiation (W m$^{-2}$), G is the ground heat flux (W m$^{-2}$), $\rho_a$ is dry air density (kg m$^{-3}$), $c_p$ is





the specific heat capacity of air (J kg$^{-1}$ K$^{-1}$), $D$ is the vapor pressure deficit (Pa), $\gamma$ is the psychrometric constant (Pa K$^{-1}$), $g_a$ is the conductance of the atmosphere, and $g_{surf}$ is surface conductance to water vapor flux (both m s$^{-1}$ but can also be expressed in flux units of mol m$^{-2}$ s$^{-1}$). $g_{surf}$ includes canopy conductance from stomatal opening, $g_c$, associated with $T$, and other conductances, often taken to be those from soil evaporation ($g_{soil}$) associated with $E_{soil}$ and plant-intercepted evaporation ($g_i$)

associated with $E_i$, the combination of which results in ecosystem-scale $E$. In brief, the biological drivers that alter $g_c$ impact $T$, but physical drivers impact both $E$ and $T$. In practice, the Penman-Monteith equation is commonly simplified because of the challenge of correctly simulating all relevant conductances (Maes et al., 2018; Priestley and Taylor, 1972).

The micrometeorological drivers of the Penman-Monteith equation vary within and across plant canopies and landscapes (Jarvis and McNaughton, 1986) as do the turbulent structures that transport water into the atmosphere by which $ET$ can be

measured using eddy covariance. Because $ET$ is commonly measured above plant canopies with eddy covariance, micrometeorological variables are commonly measured above plant canopies as well, but do not necessarily reflect micrometeorological conditions at evaporating and transpiring surfaces. For example, characteristic profiles of water vapor concentration in the atmosphere measured above the plant canopy are different from $D$ at the canopy, leaf, and soil levels (De Kauwe et al., 2017; Jarvis and McNaughton, 1986; Lin et al., 2018). Furthermore, the fundamental assumption that $D$ reflects

the difference between atmospheric water vapor pressure and saturated conditions within the leaf is challenged by studies demonstrating that leaf vapor pressure need not be saturated (Cernusak et al., 2018). Radiation, temperature, and wind speed also vary throughout plant canopies with consequences for modeling $T$ from the canopy and $E$ from the soil and other ecosystem surfaces. The space-time variability of environmental drivers within plant canopies should therefore ideally be measured or simulated to understand how they impact $E$ and $T$, and ecosystem modelers must decide if this canopy-resolved

detail is important to simulate (Medvigy et al., 2009) in diverse ecosystems (Boulet et al., 1999; Polhamus et al., 2013).

Modeling $ET$ at the ecosystem scale is challenging enough before noting that ongoing changes to the Earth system impact all of the biotic and abiotic variables that determine it. The decline in incident radiation across some regions of the world due largely to anthropogenic aerosols ('global dimming') and subsequent increase since about 1990 ('global brightening') have changed surface $R_n$ (Wild et al., 2005). The observed decrease in wind speed ('global stilling') (McVicar et al., 2012a, 2012b)

is partly due to increases in surface roughness owing to increases in $LAI$ (Vautard et al., 2010) and has decreased $g_a$, which is a function of wind speed (Campbell and Norman, 1998). Atmospheric heating changes the terms in Eq. 1 that involve temperature, namely $R_n$ (via incident longwave radiation), $\lambda$, $\gamma$, and $s$ through the Clausius-Clapeyron relation. A warming climate also increases $D$ in the absence of changes in specific humidity, but specific humidity increased across many global regions (Willett et al., 2008) resulting in complex spatial and temporal changes in $D$ (Ficklin and Novick, 2017). $g_c$ is controlled

by soil moisture availability (Porporato et al., 2004), plant hydrodynamics (Bohrer et al., 2005; Matheny et al., 2014), and environmental variables including $D$ that result in stomatal closure (Oren et al., 1999) (Fig. 2). This dependency on $D$ is predicted to become increasingly important as global temperatures continue to rise (Novick et al., 2016), but $D$ is also highly coupled to soil moisture (Zhou et al., 2019), and both depend on $ET$ itself through soil-vegetation-atmosphere coupling. Increases in atmospheric $CO_2$ concentration tend to decrease stomatal conductance at the leaf scale (Field et al., 1995) and

have been argued to decrease $g_c$ on a global scale (Gedney et al., 2006). However, elevated $CO_2$ often favors increases in $LAI$



(e.g. Ellsworth et al., 1996), thus leading to an increase in transpiring area which can support greater $g_c$. Atmospheric pollutants including ozone also impact $g_c$ with important consequences for vegetation function (Hill et al., 1969; Wittig et al., 2007). Water fluxes from the land surface impact atmospheric boundary layer processes including cloud formation, extreme temperatures, and precipitation (Gerken et al., 2018; Lemordant et al., 2016; Lemordant and Gentine, 2018), which feeds back

to land surface fluxes in ways that are inherently nonlinear and difficult to simulate (Ruddell et al., 2013). In addition to these highly nonlinear dynamics of the soil-vegetation-atmosphere system, ongoing land use and land cover changes impact vegetation structure and function with important implications for the water cycle. In brief, we need to correctly simulate how $E$ and $T$ respond to a range of biotic and abiotic variability for predictive understanding. To do so, we need to accurately measure $E$ and $T$ in the first place.

## 10  3 Measuring evaporation and transpiration

Multiple reviews and syntheses of $E$ and $T$ measurements have been written (e.g. Abtew and Melesse, 2012; Anderson et al., 2017b; Kool et al., 2014; Shuttleworth, 2007; Wang and Dickinson, 2012) and it is not our intent to reiterate them. Rather, we focus on existing and emerging approaches to partition $E$ and $T$ at the ecosystem scale on the order of tens of meters to kilometers at temporal resolutions on the order of minutes to hours. We do so to align ecosystem-scale observations of $E$ and

$T$ with satellite-based algorithms which can scale $E$ and $T$ from ecosystem to region to globe.

$ET$ is commonly approximated as the residual of the water balance at the watershed scale in hydrologic studies, but now can be measured using eddy covariance at the ecosystem scale (Wilson et al., 2001). Other approaches including scintillometry (Cammalleri et al., 2010; Hemakumara et al., 2003), surface renewal (Snyder et al., 1996), and the Bowen Ratio Energy Balance method provide important complements to eddy covariance techniques. There are multiple ways to measure ecosystem

$E$ and $T$, including leaf gas exchange, tree-level sap flow, lysimeters, soil, leaf, and canopy chambers, potometers, soil heat pulse methods, and stable and radioisotopic techniques. Ongoing efforts to synthesize measurements of ecosystem water cycle components – for example SAPFLUXNET (Poyatos et al., 2016) – are a promising approach to build understanding of different terms of the ecosystem water balance across global ecosystems. Such syntheses follow ongoing efforts to compile $ET$ measured by eddy covariance via FLUXNET and cooperating consortia (Chu et al., 2017), which synthesize half-hourly to hourly eddy

covariance flux measurements that have been used to partition $ET$ into $E$ and $T$ with mixed success.

### 3.1 Partitioning ET using half-hourly eddy covariance observations

An early attempt to partition $E$ and $T$ directly from eddy covariance measurements assumed that $ET$ is comprised solely of E in the absence of canopy photosynthesis (gross primary productivity, $GPP$) due to the coupled flux of carbon and water through plant stomata (Stoy et al., 2006). It was further assumed that $E_{soil}$ dominated $ET$ during these times, and that $E_{soil}$ could be

modeled by simulating solar radiation attenuation through grass, pine forest, and deciduous forest canopies in the Duke Forest, NC, USA. $T$ was subsequently approximated as the difference between measured $ET$ and the model for $E_{soil}$ during times when photosynthesis was active. Annual $T/ET$ values from this approach varied from 0.35 to 0.66 in the grass ecosystem (US-Dk1)





across a four year period and between 0.7 to 0.75 in the pine (US-Dk3) and hardwood (US-Dk2) forests, somewhat higher than global syntheses (Schlesinger and Jasechko, 2014), remote sensing estimates from PT-JPL (see Appendix A) for the Duke pine forest (Fig. 3), and sap flow-based measurements from the deciduous forest (Oishi et al., 2008). These discrepancies arose due in part because $E_i$ was considered negligible but can be considerable (see section 3.6). The model for $E_{soil}$ could also not be

directly validated using measurements from the forest floor alone with available observations.

An under-explored approach for partitioning $E_{soil}$ from ecosystem $ET$ uses concurrent above and below canopy eddy covariance measurements in forest and savanna ecosystems (Misson et al., 2007). Subcanopy eddy covariance measurements have proven useful for measuring below-canopy $ET$, often assumed to be comprised largely of $E_{soil}$ in ecosystems with poor understory cover (Baldocchi et al., 1997; Baldocchi and Ryu, 2011; Moore et al., 1996; Sulman et al., 2016). However, such measurements

are not yet widely adopted for $ET$ partitioning studies due to a limited understanding of their performance (Perez-Priego et al., 2017); most work to date has used below-canopy eddy covariance to partition canopy $GPP$ and soil respiration (Misson et al., 2007). Several recent studies demonstrated the additional value of concurrent below-canopy measurements for quantifying the coupling and decoupling of below- and above-canopy airspace to accurately apply the eddy covariance technique in forested ecosystems (Jocher et al., 2017, 2018; Paul-Limoges et al., 2017; Thomas et al., 2013), arguing that below-canopy eddy

covariance measurements should be more widely adopted. Other eddy covariance-based partitioning methods take a different approach and use the relationship between $T$ and $GPP$ to partition ecosystem-scale $E$ and $T$.

Scott and Biederman (2017) assumed that T is linearly related to GPP at monthly time scales over many years such that:

$$T = mrGPP \qquad (2)$$

where m is the inverse of the marginal water use efficiency ($\Delta ET/\Delta GPP$) and $r$, the ratio between the inverse of the

transpirational water use efficiency ($\Delta T/\Delta GPP$) and the marginal ecosystem water use efficiency, is assumed to be unity. It follows that the intercept of the relationship $ET = mGPP + E'$ is an estimate of average monthly $E$. This approach is favored in semi-arid ecosystems where there is a close coupling of $ET$ and $GPP$ and when $E$ is a considerable fraction of $ET$.

Several recently developed methods for partitioning eddy covariance-measured $ET$ are based on the optimality theory assumption that plants minimize water loss per unit $CO_2$ gain (e.g. Hari et al., 2000; Katul et al., 2009; Medlyn et al., 2011;

Schymanski et al., 2007). An outcome of this approach is that $WUE$ scales with $D^{0.5}$ from which a relationship between $GPP$ and $T$ can be derived (Katul et al., 2009). Berkelhammer et al. (2016) also noted that $ET$ follows a linear relationship to $GPP \times D^{0.5}$ and further assumed that the $T/ET$ ratio intermittently approaches 1. They then separated $ET$ measurements from eddy covariance into $GPP$ classes for which a minimum $ET$, $min(ET)|_{GPP}$, can be defined. $T/ET$ can then be calculated using:

$$T/ET = \frac{ET}{min(ET)|_{GPP}} \qquad (3)$$

Applying this approach to different forests revealed considerable synoptic scale variability in $T/ET$ that was dampened at seasonal time scales and compared well against isotopic approaches.

Zhou et al. (2016) built upon earlier work (Zhou et al., 2014) and assumed an ecosystem has an *actual* underlying water use efficiency ($uWUE_a$) which is maximal or reaches its *potential* underlying water use efficiency ($uWUE_p$) when $T/ET$ approaches unity. $T/ET$ can thus be calculated from the ratio of actual to potential $uWUE$ using optimality assumptions for both:



$$uWUE_p = \frac{GPP\sqrt{D}}{T} \qquad\qquad\qquad\qquad (4)$$

and

$$uWUE_a = \frac{GPP\sqrt{D}}{ET} \ . \qquad\qquad\qquad\qquad (5)$$

Again, assuming that $T/ET$ intermittently approaches one in sub-daily eddy covariance measurements, the $uWUE_p$ can be
estimated empirically using 95th quantile regression to find the upper boundary of the relationship between measured $ET$ and
$GPP \times D^{0.5}$. $uWUE_a$ can be calculated using eddy covariance observations, and $T$ estimates using this approach compare well
against independent sap flow measurements (Zhou et al., 2018). A semi-empirical model based on the $uWUE$ concept by Boese
et al. (2017) included radiation and was able to outperform the Zhou et al. (2016) approach, on average, consistent with the
notion that $T$ is also driven by radiation (Eq. 1) (Pieruschka et al., 2010). It is important to note when applying $WUE$-based
approaches that there are important discrepancies between $WUE$ measurements at leaf and canopy scales that still need to be
resolved (Medlyn et al., 2017; Medrano et al., 2015).

In a more sophisticated attempt to partition $ET$ utilizing optimality theory, (Perez-Priego et al., 2018) utilized a big-leaf canopy
model where parameters were optimized using half-hourly data in five-day windows. Uniquely, the marginal carbon cost of
water was factored into the cost function during parameter estimation, so the parameters for each five-day window maximized
the fit between modeled and observed $GPP$ and also minimized water loss per carbon gain. $T$ was then calculated using $g_c$
from the model, and $E$ was calculated as the residual ($ET - T$).

A modified (in this case binned) parameter optimization approach was used by Li et al. (2019), which follows the model
proposed by Lin et al. (2018):

$$g_{surf} = g_0 + g_1 \frac{GPP}{D_l^m} \qquad\qquad\qquad\qquad (6)$$

Here, $g_0$ (assumed to correspond to soil conductance), $g_1$ (assumed to correspond to vegetation conductance), and $m$ are
optimized parameters, $D_l$ is the inferred leaf level $D$, and $g_{surf}$ is estimated by inverting Eq. 1 (and is assumed to represent
ecosystem conductance). Rather than optimizing using a moving window over time, data were binned using independent soil
moisture data associated with the eddy covariance site, with $g_0$, $g_1$, and $m$ optimized in each bin to account for changes due to
water limitations. Partitioning was then calculated as:

$$\frac{T}{ET} = \frac{g_1}{g_{surf}} \qquad\qquad\qquad\qquad (7)$$

and

$$\frac{E}{ET} = \frac{g_0}{g_{surf}}. \qquad\qquad\qquad\qquad (8)$$

The Perez-Priego et al. (2018) and Li et al. (2019) methods both circumvent the assumption that $T/ET$ approaches unity at
some periods by estimating ecosystem conductances. The Transpiration Estimation Algorithm (TEA) from Nelson et al. (2018)
utilizes a non-parametric model and thereby further limits assumptions made about how the ecosystem functions. However,
TEA must make the assumption that $T/ET$ approaches one, which it does by filtering periods when the surface is likely to be
wet. In a validation study which utilized model output as synthetic eddy covariance datasets where $E$ and $T$ are known, TEA





was able to predict $T/ET$ patterns in both space and time but showed a sensitivity to the minimum modelled $E$. Overall, TEA was able to predict temporal patterns of $T$ across three different ecosystem models and provides an important basis for comparison because the model for $T$ is agnostic to underlying ecosystem function.

## 3.2 Advanced algorithms for partitioning eddy covariance data

Scanlon and Kustas (2010) (see also Scanlon and Sahu, 2008) developed a partitioning approach for $E$ and $T$ based on the notion that atmospheric eddies transporting $CO_2$ and water vapor from stomatal processes ($T$ and $GPP$) and non-stomatal processes ($E$ and respiration) independently follow flux variance similarity as predicted by Monin Obukhov Similarity Theory. In brief, there are two end-member scenarios for a parcel of air transported from a surface: one without stomata, and one with stomata. An eddy transported away from a surface that is respiring $CO_2$ and evaporating water through pathways other than

stomata will have deviations from mean $CO_2$ mixing ratio ($c'$) and water vapor mixing ratio ($q'$) that are positively correlated. An eddy of air transported by a surface with stomata will have a negative relationship between $c'$ and $q'$ that is determined by a unique canopy $WUE$. $WUE$ is thereby used to establish a functional relationship between the variance of $CO_2$ due to stomatal uptake ($\sigma_{cr}$) and the correlation between stomatal and non-stomatal $CO_2$ exchange processes ($\rho_{cp,cr}$). Subsequently, $ET$ can be partitioned into its $T$ and $E$ components by matching the observed correlation of $q'$ and $c'$ ($\rho_{q,c}$) to the corresponding value of

$\rho_{cp,cr}$ (Scanlon and Sahu, 2008). The original approach applied wavelet filtering to remove large-scale atmospheric effects that impact the validity of underlying flux-variance relationships and was shown to realistically reproduce $T/ET$ relationships over the growing period of a corn (maize) crop (Scanlon and Kustas, 2012).

Subsequent work by Skaggs et al. (2018) noted that there is an algebraic solution to terms that had previously been solved using optimization (namely $\sigma_{cr}$ and $\rho_{cp,cr}$, (Palatella et al., 2014)) and created a Python module, fluxpart, to calculate $E$

and $T$ using the flux variance similarity approach. The original flux variance similarity approach used a $WUE$ formulation following Campbell and Norman (1998); fluxpart allows $WUE$ to vary as a function of $D$ or take a constant value. $WUE$ varies throughout the canopy and in response to other environmental conditions and using high frequency rather than leaf-level observations to estimate it results in uncertainties (Perez-Priego et al., 2018) which can be addressed in part by using outgoing longwave radiative flux density observations to estimate canopy temperature (Klosterhalfen et al., 2018, 2019). A careful

comparison of flux variance partitioning results against fluxes simulated by large eddy simulation revealed that it yields better results during fair weather (i.e. not during or after precipitation events) and with a developed plant canopy with clear separation of $CO_2$ and $H_2O$ sources and sinks. It is also possible to separate $E$ and $T$ using conditional sampling of turbulent eddies (Thomas et al., 2008); performance of the conditional sampling method is a function of canopy height and leaf area index and performance of the flux variance similarity method is related to the ratio between sensor height and canopy height

(Klosterhalfen et al., 2018), suggesting that different methods may deliver better results in different ecosystems

It should also be noted that flux variance similarity can be used directly with half-hourly flux data if the wavelet filtering step is unnecessary, but in practice high-frequency eddy covariance data are required because the necessary terms are rarely computed and saved. Of course, all eddy covariance-based $ET$ partitioning approaches need to critique the energy balance closure of the observations (Leuning et al., 2012; Stoy et al., 2013; Wohlfahrt et al., 2009) especially in closed-path eddy



covariance systems which are prone to water vapor attenuation in the inlet tube (Fratini et al., 2012; Mammarella et al., 2009). Eddy covariance-based approaches to partition $E$ and $T$ can be complemented by other new approaches to measure or estimate $T$ at commensurate spatial and temporal scales.

### 3.3 Solar-induced fluorescence (SIF)

$GPP$ and $T$ are coupled through stomatal function, and studies of $GPP$ have recently been revolutionized by space and ground-based observations of solar-induced fluorescence (SIF) (Frankenberg et al., 2011; Gu et al., 2018; Köhler et al., 2018; Meroni et al., 2009), the process by which some of the incoming radiation that is absorbed by the leaf is re-emitted by chlorophyll. SIF emission is related to the light reactions of photosynthesis but $GPP$ estimation also requires information on the dark reactions and stomatal conductance such that the remote sensing community is currently challenged by how to use SIF to

estimate $GPP$. New studies also propose that SIF might be used to monitor $T$, possibly in combination with surface temperature measurements, acknowledging the close link between $GPP$ and $T$ due to their joint dependence on stomatal conductance and common meteorological and environmental drivers (Alemohammad et al., 2017; Damm et al., 2018; Lu et al., 2018; Pagán et al., 2019; Shan et al., 2019).

While SIF is related to the electron transport rate (Zhang et al., 2014), $T$ primarily depends on stomatal conductance such that

SIF and $T$ are linked empirically but not mechanistically. This link is expected if $GPP$ and $T$ are tightly coupled. SIF has also been proposed to predict the ecosystem-scale water use efficiency (defined here as $EWUE = GPP / T$) (Lu et al., 2018), a critical component of many of the $E$ and $T$ partitioning algorithms based on eddy covariance $ET$ measurements described above. Shan et al. (2019) showed that $T$ can be empirically derived from SIF in forest and crop ecosystems with explained variance ranging from 0.57 to 0.83 and to a lesser extent in grasslands with explained variance between 0.13 and 0.22. The

authors suggested that the decoupling between $GPP$ and $T$ during water stress episodes hampered the use of SIF to predict $T$, particularly in grasslands, noting that $T$ can occur without $GPP$ under periods of plant stress (Bunce, 1988; De Kauwe et al., 2019). There is a strong empirical link between the ratio of $T$ over potential evaporation and the ratio of SIF over $PAR$, and the relationship depends on the atmospheric demand for water, with larger transpiration for the same SIF when potential evaporation is higher Alemohammad et al., 2017; Damm et al., 2018; Lu et al., 2018; Pagán et al., 2019; Shan et al., 2019).

These ratios vary with assumptions regarding the potential evaporation calculation as well (Fisher et al., 2010). SIF can be measured at unprecedented spatial and temporal scales (Köhler et al., 2018) including the scale of the eddy covariance flux footprint (Gu et al., 2018), and this information can in turn be incorporated into remote sensing-based approaches for estimating $ET$ using remote sensing platforms (see Appendix A) following additional mechanistic studies of its relationship with $T$.

### 3.4 Carbonyl sulfide (COS) flux

Other approaches to estimate GPP use independent tracers such as carbonyl sulfide (COS). When plants open their stomata to take up $CO_2$ for photosynthesis, they also take up COS (Campbell et al., 2008), a trace gas present in the atmosphere at a global average mole fraction of $\sim 500$ ppt (Montzka et al., 2007). The leaf-scale uptake of COS, $F_{COS}$ (pmol m-2 s-1), can be calculated using





$$F_{COS} = -C_{COS} \left( \frac{1}{g_b} + \frac{1}{g_{s,COS}} + \frac{1}{g_i} \right)^{-1}. \tag{9}$$

where $C_{COS}$ (pmol mol$^{-1}$) is mole fraction of COS and $g_b$, $g_{s,COS}$, and $g_i$ represent the leaf-scale boundary layer, stomatal and internal conductances (mol m$^{-2}$ s$^{-1}$) to COS exchange (Sandoval-Soto et al., 2005; Wohlfahrt et al., 2012). The latter lumps together the mesophyll conductance and the biochemical "conductance" imposed by the reaction rate of carbonic anhydrase, the enzyme ultimately responsible for the destruction of COS (Wehr et al., 2017). Equation 9 also makes the common assumption that, because the carbonic anhydrase is highly efficient in catalyzing COS, the COS mole fraction at the diffusion endpoint is effectively zero (Protoschill-Krebs et al., 1996). Provided appropriate vertical integration over the canopy is made, Eq. 9 can be used to describe canopy-scale $F_{COS}$ (Wehr et al., 2017).

Because COS and $CO_2$ share a similar diffusion pathway into leaves and because the leaf exchange of COS is generally unidirectional, COS has been suggested (Sandoval-Soto et al., 2005; Seibt et al., 2010; Wohlfahrt et al., 2012) and demonstrated (Wehr et al., 2017; Yang et al., 2018) to present an independent proxy for estimating *GPP*. Motivated by the common boundary layer and stomatal conductances, there has been recent interest in using measurements of the COS exchange to estimate the canopy stomatal conductance to water vapor and by extension *T* (Asaf et al., 2013; Wehr et al., 2017; Yang et al., 2018). Solving for $g_{s,COS}$ from Eq. 9 requires measurements of $F_{COS}$ (e.g. by means of eddy covariance; (Gerdel et al., 2017)) and $C_{COS}$, while $g_b$ and $g_i$ are typically estimated based on models.

With $g_s$ (and by canoy scaling $g_c$) determined this way, and an estimate of the aerodynamic conductance (the canopy-analog to the leaf boundary-layer conductance, Eq. 1), T may be derived by multiplication with the canopy-integrated leaf-to-air water vapor gradient. The first and to-date only study to attempt this was conducted by Wehr et al. (2017), who demonstrated excellent correspondence with $g_c$ estimated from *ET* measurements in a temperate deciduous forest. While stomata dominated the limitation of the COS uptake during most of the day, co-limitation by the biochemical "conductance" imposed by carbonic anhydrase was observed around noon. This finding is consistent with leaf-level studies by Sun et al. (2018) and suggests that $g_i$ in Eq. 9 may not generally be negligible, even though Yang et al. (2018) found the bulk surface conductance of COS (i.e. all conductance terms in Eq. 9 lumped together) to correspond well with the surface conductance for water vapor inferred from *ET*. As soils may both emit and take up COS, ecosystem-scale COS flux measurements need to account for any soil exchange, even though typically the soil contribution is small (Maseyk et al., 2014; Whelan et al., 2018). One notable exception for larger soil $F_{COS}$ fluxes occurs in some agricultural systems (Whelan et al., 2015), due in part to the relationship of $F_{COS}$ with soil nitrogen (Kaisermann et al., 2018). Clearly, further studies are required in order to establish whether the complexities of and uncertainties associated with inferring gs from Eq. 9 and non-stomatal fluxes make COS observations a sensible independent alternative for estimating canopy *T*.

## 3.5 Advances in thermal imaging

Thermal remote sensing measures the radiometric surface temperature following the Stefan-Boltzmann Law. *ET* can be estimated using thermal remote sensing by applying an ecosystem energy balance residual approach: $\lambda E = R_n - G - H$ (Norman et al., 1995). Quantifying the available energy term ($R_n - G$) is difficult from space and the radiometric surface temperature





measured by infrared sensors is different from the aerodynamic surface temperature that gives rise to sensible heat flux ($H$) (Kustas and Norman, 1996). Despite these challenges, thermal remote sensing for $ET$ has been widely used with multiple satellite platforms including Landsat, MODIS, Sentinel, and GOES (Anderson et al., 2012; Fisher et al., 2017; Semmens et al., 2016). One of NASA's newest missions is ECOSTRESS mounted on the International Space Station, which produces

thermally-derived $ET$ at 70 m resolution with diurnal sampling, (Fisher et al., 2017).

Advances in thermal imaging (thermography) have made it possible to make radiometric surface temperature observations at increasingly fine spatial and temporal resolutions (Jones, 2004) – on the order of millimeters or less. Thermography has been used to estimate $E_{soil}$ (Haghighi and Or, 2015; Nachshon et al., 2011; Shahraeeni and Or, 2010) and $T$ from plant canopies (Jones, 1999; Jones et al., 2002), often in agricultural settings (Ishimwe et al., 2014; Vadivambal and Jayas, 2010). Researchers

are increasingly using tower and UAV-mounted thermal cameras to measure the temperatures of different ecosystem components at high temporal and spatial resolution (Hoffmann et al., 2016; Pau et al., 2018), which could revolutionize the measurement of $T$ from plant canopies (Aubrecht et al., 2016) or even individual leaves in a field setting (Page et al., 2018). Such measurements need to consider simultaneous $E_i$ and $T$ from wet leaf surfaces.

### 3.6 The challenges of measuring evaporation from canopy interception

$E_i$ from wet canopies can return 15-30% or more of incident precipitation back into the atmosphere annually (Crockford and Richardson, 2000). Although interception has been studied over a century, the underlying physical processes, atmospheric conditions, and canopy characteristics that affect it are poorly understood (van Dijk et al., 2015). Accurately estimating $E_i$ from wet canopies is critical for the proper simulation of interception loss (Pereira et al., 2016). However, $E_i$ predicted by the Penman–Monteith equation (Eq. 1) during rainfall is often a factor of two or more smaller than the $E_i$ derived from canopy

water budget measurements (Schellekens et al., 1999). A recent study using detailed meteorological measurements from a flux tower indicates that the underestimated $E_i$ by the Penman-Monteith equation might be attributed to the failure in accounting for the downward sensible heat flux and heat release from canopy biomass as they can be major energy sources for wet-canopy evaporation (Cisneros Vaca et al., 2018). Storm characteristics (e.g., amount, storm duration, and intensity) and canopy structural information (e.g., canopy openness, canopy storage capacity) are all important parameters for modeling $E_i$ (van Dijk

et al., 2015; Linhoss and Siegert, 2016; Wohlfahrt et al., 2006). To partition total $ET$ into $T$, $E_{soil}$, and $E_i$, it is necessary to simulate the dynamics of canopy wetness before, during, and after each storm so that models can be applied to the dry and wet portions of the canopy, respectively (Liu et al., 1998), a process that can be implemented using a running canopy water balance model (Liu, 2001; Rutter et al., 1971; Wang et al., 2007). Understanding the sources of water is therefore useful for quantifying differences among $T$, $E_{soil}$, and $E_i$, and information from water isotopes can be helpful to do so.

### 3.7 Isotopic approaches

The hydrogen and oxygen atoms of water molecules exist in multiple isotopic forms, including $^2H$ and $^{18}O$, which are stable in the environment and can be used to trace the movement of water through hydrologic pathways (Bowen and Good, 2015; Gat, 1996; Good et al., 2015; Kendall and McDonnell, 2012). Because heavier atoms preferentially remain in the more

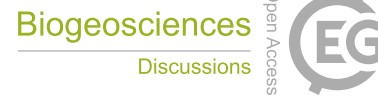



condensed form during phase change, evaporation enriches soils in $^2$H and $^{18}$O (Allison and Barnes, 1983), while root water uptake typically removes water from the soil without changing its isotope ratio (Flanagan and Ehleringer, 1991). This difference in the isotope ratio, $R = [^2H]/[^1H]$ or $[^{18}O]/[^{16}O]$, of soil evaporation compared with the isotope ratio of water moving through plants is the basis for isotopic partitioning of evapotranspiration. If evapotranspiration consists of two components, $E$

and $T$, with distinct isotopic composition: $R_E$ for soil evaporation and $R_T$ for plant transpiration, then the bulk flux, $R_{ET}$, can be incorporated into a simple mass balance of the rate isotope (i.e.: $R_{ET}ET = R_E E + R_T T$), which can be rearranged as (Yakir and Sternberg, 2000):

$$\frac{T}{ET} = \frac{R_{ET} - R_E}{R_T - R_E}. \tag{10}$$

Thus, knowledge of the isotopic ratio of each flux component, $R_E$ and $R_T$, as well as the total bulk flux isotope ratio, $R_{ET}$, is

sufficient to estimate the fraction that passes through plants.

Techniques to measure the $R_{ET}$ have diversified since the widespread deployment of laser-based integrated cavity output spectroscopy (ICOS) systems, which are currently monitoring atmospheric stable isotope ratios, $R_A$, at a wide number of sites (Wei et al., 2019; Welp et al., 2012). Vertical profiles and high frequency measurements of $R_A$ are used to determine $R_{ET}$ though a number of different methods, all of which are associated with potentially large uncertainty (Griffis et al., 2005, 2010;

Keeling, 1958). Propagation of uncertainties through Eq. 10 demonstrates that errors in $R_{ET}$, $R_T$, and $R_E$, as well as differences between $R_E$ and $R_T$, strongly influence the final partitioning estimate (Good et al., 2014; Phillips and Gregg, 2001). The isotopic approach becomes uninformative as $R_E$ approaches $R_T$. Furthermore, as $E_i$ adds another source term to the isotope mass balance, Eq. (10) can be implemented over short periods only when the canopy is dry. If $E_i$ is incorporated as a third source, its magnitude and isotope ratio must be specified, and these assumptions can strongly influence any final isotope based

partitioning estimates (Coenders-Gerrits et al., 2014; Schlesinger and Jasechko, 2014).

The value of $R_E$ is derived from the soil water isotope ratio, $R_S$, as well as the temperature and humidity conditions under which evaporation happened (Craig and Gordon, 1965). Destructive extraction of water from soil cores can be used to estimate $R_S$, though recent studies have highlighted discrepancies between methodologies (Orlowski et al., 2016a, 2016b). *In situ* monitoring of $R_S$ obtained by pumping soil vapor through ICOS systems has been demonstrated (Gaj et al., 2016; Oerter et al.,

2016; Volkmann and Weiler, 2014), and recently applied to $ET$ partitioning to provide continuous updates on soil isotope ratios (Quade et al., 2019). Eddy covariance measurements of $^2$H and $^{18}$O are now possible (Braden-Behrens et al., 2019). However, identifying $R_S$ remains challenging, and the bulk soil moisture composition (Mathieu and Bariac, 1996; Soderberg et al., 2013), depth (Braud et al., 2005), and soil physical composition (Oerter et al., 2014) where evaporation occurs can alter the $R_S$ to $R_E$ relationship.

If water entering the plant is isotopically the same as transpired water, known as the isotopic steady-state assumption, then $R_T$ = $R_S$. However, preferential uptake at the root-soil interface, differences between plant internal water pools in time, and mixing along the water pathways within plants, will invalidate the steady-state assumption (Farquhar and Cernusak, 2005; Ogée et al., 2007). Finally, variability between and within plant species and plant/soil microclimates of an ecosystem will move the system away from the simple two-source model used in Eq. 10. Accurate knowledge of the isotope ratio within various water reservoirs



of a landscape, including the planetary boundary layer (Noone et al., 2013), and how these translate into distinct water fluxes is required to advanced isotope-based partitioning approaches.

## 4 Critiquing the assumptions of *ET* partitioning methods

### 4.1 Do ecosystems exhibit optimal responses to *D*?

Many *WUE*-based approaches for partitioning *E* and *T* (sections 3.1 & 3.2) hinge on the notion that canopy conductance ($g_c$) follows an optimal response to *D*. Recent data-driven studies have argued that canopy conductance measured using eddy covariance is 'slightly suboptimal' and scales with $D^{0.55}$ rather than $D^{0.5}$ (Lin et al., 2018) or is 'nearly optimal' and scales with $GPP \times D^{0.55}$ (Zhou et al., 2015). Here, we test the assumption that plant canopies exhibit optimal responses to *D* by assuming that it serves as a *constraint* on *EWUE* following an implication of optimality theory that one minus the ratio of leaf-internal

$CO_2$ ($c_i$) to atmospheric $CO_2$ ($c_a$) ($1-c_i/c_a$) also scales with $D^{0.5}$ (see Eq. 18 in (Katul et al., 2009)). From the definition of *EWUE* and expanding *GPP* and *T* using Fick's Law it follows:

$$EWUE = \frac{GPP}{T} = \frac{g_c \varepsilon c_a \left(1 - \frac{c_i}{c_a}\right)}{g_c D}. \tag{11}$$

In this equation, $g_c$ cancels and $\varepsilon$ is the relative diffusivity of $H_2O$ and $CO_2$ molecules. If ($1 - c_i/c_a$) scales with $D^{0.5}$, eddy covariance-measured *EWUE* should therefore scale with $D^{-0.5}$ if it can be assumed that measured *ET* approaches *T*. We test

this notion using micrometeorological and eddy-covariance data from 240 sites that include ecosystem type and ecosystem energy balance measurements in the LaThuile FLUXNET database following Stoy et al. (2013). Specifically, we use a boundary line analysis commonly used in studies of leaf and canopy conductance (Schäfer, 2011), taking the mean of the upper 95% of eddy covariance *EWUE* observations in 0.3 kPa bins of *D* and fit an exponential model to these observations using nonlinear least squares (Fig. 4a) (rather than fitting a linear model following log transformation for values that approach zero).

Using this approach, a mean (± standard deviation) exponential term of -0.53 ± 0.17 from the 240 sites is calculated (Fig. 4b), which is not significantly different from -0.5 using a one-sample t-test. Repeating this analysis with the FLUXNET2015 dataset reveals a mean exponential term of -0.49 ± 0.15, which is likewise not different from -0.5.

Land surface models struggle to simulate this emergent property of ecosystems. Models for the ecosystems shown in Fig. 1 tend to dramatically over-predict the magnitude of the exponential term with a mean value of -2.9 (Table 2). The exponential

term of the BEPS model was -0.54 ± 0.06, similar to observations. Combined, these results suggest that an optimal canopy response to *D* may be a reasonable assumption despite the challenges of leaf-to-ecosystem scaling, but the considerable variability of the calculated exponential terms suggest that more research is necessary to understand conditions under which optimality is a reasonable assumption when it is not. The discrepancy in calculated exponential terms between measurements and models further emphasize the importance of improved carbon and water coupling in ecosystem models.



## 4.2 Does *T*/*ET* approach unity?

Also central to many *E* and *T* partitioning approaches is the notion that *T*/*ET* intermittently approaches 1 (Berkelhammer et al., 2016; Nelson et al., 2018; Zhou et al., 2016), as suggested by some modeling analyses (Wei et al., 2018). This assumption was critiqued by Perez-Priego et al. (2018), who demonstrated that *T*/*ET* was rarely greater than 0.8 in a Mediterranean

ecosystem, even during dry periods when surface soil moisture was less than 0.2 $m^3$ $m^{-3}$, and that *E* scaled with $time^{-0.5}$ following (Brutsaert, 2014) [see also Boese et al. (2018)]. These findings of a sustained evaporation component and non-zero *E*/*ET* even during dry conditions were also supported by lysimeter measurements in a semiarid grassland (Moran et al., 2009) and partly confirmed by a recent study based on isotopes on shrubs and steppe ecosystem (Wang et al., 2018). The maximum daily *T*/*ET* found by Scanlon and Kustas (2012) in a maize agroecosystem was also about 0.8, but Rana et al. (2018) found

daily values that intermittently exceeded 0.9 in wheat and fava bean fields and multi-method comparisons suggest that *T*/*ET* approaches 0.85 (Rafi et al., 2019). Anderson et al. (2017a) found that *T*/*ET* routinely exceeded 0.9 in sugarcane, with maximum daily values above 0.95. We can critique the notion that *T*/*ET* approaches 1 by applying the flux variance similarity partitioning approach to a wheat canopy from central Montana, USA measured by Vick et al. (2016). Wheat has a characteristically high surface conductance (Bonan, 2008) and approaches an ideal transpiring surface during the main growth

period (Bonan, 2008; Priestley and Taylor, 1972). The dryland wheat crops studied here draw water from depth such that surface soils are often dry (Vick et al., 2016), minimizing $E_{soil}$. Applying the flux variance similarity method of Scanlon and Kustas (2010) to the wheat crop suggests that *T*/*ET* frequently exceeds 0.95 during daytime periods when the algorithm converges (Fig. 5a). Repeating this analysis for a winter wheat crop near Sun River, Montana, USA using the flux variance similarity algorithm of Skaggs et al. (2018) confirms this finding with an even higher proportion of *T*/*ET* values (20%) that

exceed 0.95. *T*/*ET* however exceeded 0.95 in less than 2% of measurements using the approach of Perez-Priego et al. (2018) in a Mediterranean savanna ecosystem (Fig. 6). These observations suggest that the notion that *T*/*ET* approaches 1 is a good assumption in some ecosystems but not others, with implications for flux partitioning by the methods that rely on this assumption.

## 5 Research imperatives

Few field experiments – if any – have sought to constrain ecosystem *E* and *T* estimates using multiple observations to quantify their response to environmental variability and to test the assumptions of partitioning approaches (Perez-Priego et al., 2017, 2018). Those that do note large discrepancies in *T*/*ET* estimates from different techniques (Quade et al., 2019). Despite these challenges, a multi-measurement approach is necessary to understand different ecosystem water flux terms (Li et al., 2018), but most multi-method ecosystem-scale experiments using eddy covariance measurements seek to constrain the carbon cycle

rather than the water cycle to which it is coupled (Hanson et al., 2004; Williams et al., 2009). Here, we outline the basics of an ecosystem-scale experiment designed to address uncertainties in *E* and *T* measurements.

It would be best to introduce such an experiment in an ecosystem with relatively simple species distribution and clear separation of above and below canopy *E* and *T* sources (Klosterhalfen et al., 2019; Williams et al., 2004) before addressing





more complex ecosystems with multiple canopy layers that result in within-canopy flows that are more complex (Fu et al., 2018; Santos et al., 2016). Observations should occur on time scales commensurate with satellite remote sensing overpasses; the half-hourly time step used in most eddy covariance observations is likely sufficient to approximate conditions captured by polar-orbiting satellites. For example, MODIS has a 10:30 overpass time for TERRA and 13:30 overpass for AQUA; GOME2

makes SIF observations in the morning, while OCO-2 flies over at 13:30. There will be more opportunities to study diurnal patterns with the forthcoming OCO-3 and Geostationary Carbon Cycle Observatory (GeoCarb) and under-explored opportunities to apply geostationary satellites like GOES to $ET$ partitioning (Bradley et al., 2010), which compromises spatial resolution from the distant geostationary orbit for temporal resolution on the order of minutes. A short time step measurement is possible for chambers, lysimeters, and sapflux measurements but not some isotopic approaches (Fig. 7). Critically,

thermography, SIF, and OCS flux can also be measured at these time scales. An ideal $E$ and $T$ partitioning experiment would make them both above and below plant canopies, in conjunction with below-canopy eddy covariance, to isolate $E_{soil}$. Any measurement strategy should be cognizant of atmospheric boundary layer dynamics, and atmospheric profiles of temperature and humidity from radiosondes are necessary to apply atmospheric boundary layer-based approaches that have the emerging potential for $E$ and $T$ partitioning like evapotranspiration based on equilibrated relative humidity (ETRHEQ, Appendix B)

(Rigden et al., 2018; Rigden and Salvucci, 2015; Salvucci and Gentine, 2013). These can arise from the twice-daily global radiosonde network if stations are nearby or locally-launched radiosondes to capture the diurnal behavior of the atmospheric boundary layer. For full water balance accounting, observations of drainage from the rooting zone using drainage lysimeters, soil moisture at multiple soil levels, the flow of water down plant stems (stem flow), leaf wetness sensors, and of course multiple precipitation gages are required. Such a multi-measurement approach would also create an opportunity to compare

the performance of emerging technologies like distributed temperature sensing from fiber optic cables (Schilperoort et al., 2018), modeling cosmic ray neutron fields for soil water source estimation (Andreasen et al., 2016), and Global Navigation Satellite System Reflectometry (GNSS-R) for soil moisture estimation (Zribi et al., 2018). It remains difficult to assimilate $E$ and $T$ measurements into models using conventional data assimilation techniques because observations may contain substantial bias error yet still provide valuable information (Williams et al., 2009). Emerging approaches from machine learning in the

earth and environmental sciences may therefore be particularly useful for combining the best information from different measurement techniques into a mass and energy conserving model of the surface-atmosphere exchange of water (Reichstein et al., 2019). Regardless of the specifics of the approach, we advocate more investment into the study of $ET$ ("green water") given its central importance in provisioning resources to an increasingly resource-scarce planet (Schyns et al., 2019).

## 6 Conclusion

New measurement techniques and analytical approaches for partitioning $E$ and $T$ at the ecosystem scale provide critical opportunities to improve our understanding of the global water cycle. Ecosystem scale experiments that measure $E$ and $T$ using multiple approaches are needed to understand how $E$ and $T$ respond differently to climate variability and change across different global ecosystems, and also to critique the assumptions made by $ET$ partitioning approaches. These observations can




then further improve satellite predictions to improve confidence in our understanding of the global water cycle and how it responds to ongoing changes in climate, atmospheric composition, and human demands for water.

## A1 Appendix A

For completeness we briefly describe common algorithms used by remote sensing platforms for estimating $E$, $T$, and $ET$, noting
that additional approaches exist and are under development (El Masri et al., 2019). Many widely-used algorithms including SEBAL (Bastiaanssen et al., 1998), METRIC (Allen et al., 2007; Su, 2002), and SEBS (Su, 2002) use an energy balance approach that does not explicitly seek to separate $T$ from $E$, but remain highly valuable for water resource management and hydrology.

### A1.1 PT-JPL

The Priestley-Taylor Jet Propulsion Lab (PT-JPL) global ET remote sensing retrieval algorithm (Fisher et al., 2008) is based on the potential evapotranspiration ($PET$) formulation of Priestley and Taylor (1972), which replaces the adiabatic terms in Eq. 1 with a parameter, $\alpha_{PT}$, that takes a value of 1.26 under ideal evaporating conditions

$$PET = \alpha_{PT} \frac{s(R_n - G)}{s + \gamma} \tag{A1}$$

To reduce $PET$ to $ET$, (Fisher et al., 2008) introduced ecophysiological constraint functions (f-functions, unitless multipliers
between zero and one) following (Jarvis, 1976). These are based on $D$, relative humidity ($RH$), the normalized difference vegetation index (NDVI) and the soil-adjusted vegetation index (SAVI) (Huete, 1988). PT-JPL calculates $T$, $E_{soil}$ and $E_i$ explicitly using

$$ET = T + E_{soil} + E_i \tag{A2}$$

$$T = (1 - f_{wet}) f_g f_T f_M \alpha_{PT} \frac{s(R_{nc} - G)}{s + \gamma} \tag{A3}$$

$$E_{soil} = (f_{wet} + f_{SM}(1 - f_{wet})) \alpha_{PT} \frac{s(R_{ns} - G)}{s + \gamma} \tag{A4}$$

$$E_i = f_{wet} \alpha_{PT} \frac{s(R_{nc} - G)}{s + \gamma} \qquad + \tag{A5}$$

Where $f_{wet}$ is relative surface wetness ($RH^4$), $f_g$ is green canopy fraction, $f_T$ is a plant temperature constraint, $f_M$ is a plant moisture constraint, and $f_{SM}$ is a soil moisture constraint. $R_{nc}$ and $R_{ns}$ are net radiation absorbed by canopy and soil respectively. PT-JPL has been tested against measured $ET$ from hundreds of FLUXNET sites worldwide, with a monthly average $r^2$ of 0.90
across all sites and a slope/bias of 1.07 using *in situ* data (Fisher et al., 2008, 2009). The PT-JPL model forms the core $ET$ retrieval algorithm in the ECOSTRESS mission (Fisher et al., 2017; Hulley et al., 2017) on board the International Space Station. New applications of the PT-JPL algorithm have included canopy indices derived from CubeSats (Aragon et al., 2018).



### A1.2 PM-MOD16

The PM-MOD16 algorithm estimates $ET$ on eight-day intervals at 1 km$^2$ pixels across the global terrestrial surface using MODIS observations following (Mu et al., 2011) using Eq. (A2). The PM-MOD16 algorithm follows the Penman-Monteith model (Eq. 1) rather than the Priestley-Taylor model (Eq. A1) by modeling conductance terms (or resistance terms as the inverse of the conductance terms in equation 1) rather than including f-functions as in the PT-JPL algorithm. It explains ca. 86% of the variability in eddy covariance-observed $ET$ from 46 sites in North America (Mu et al., 2011).

### A1.3 GLEAM

The Global Land Evaporation Amsterdam Model (GLEAM) (Miralles et al., 2011a, 2011b) also uses a Priestley-Taylor approach (eq. A1) to estimate $ET$. It employs the Gash (1979) analytical model for canopy rainfall interception and semi-empirical stress functions that vary between 0 and 1 (similar to the *f*-functions of the PT-JPL model) to reduce PET to T for canopies with different characteristics. For T, this stress function is calculated based on the content of water in vegetation and root zone. The former is approximated based on microwave vegetation optical depth and latter is calculated using a multilayer soil model driven by observations of precipitation and updated through assimilation of microwave surface soil moisture. Validation studies against eddy-covariance data at daily time scales show average correlations typically ranging from 0.81–0.86 (Martens et al., 2017).

### A1.4 DTD

The Dual-Temperature Difference (DTD) model follows the notion that diurnal changes in air and radiometric surface temperatures are related to surface-atmosphere heat flux (Norman et al., 2000). It has since been applied to MODIS observations to estimate $ET$ (Guzinski et al., 2013) and to partition $E$ and $T$ using a Priestly-Taylor scheme described in Song et al., (2018). DTD estimates of $E$ and $T$ compared well to estimates derived using the flux variance similarity algorithm of Skaggs et al. (2018).

### A1.5 ALEXI/DisALEXI

Atmosphere-Land Exchange Inverse (ALEXI) is a multi-scale surface energy balance modeling system, building on the two-source energy balance (TSEB) land-surface representation of Norman et al. )1995). The TSEB partitions the composite radiometric surface temperature, $T_{rad}$, into soil and canopy temperatures, $T_s$ and $T_c$, based on the local vegetation cover fraction apparent at the thermal sensor view angle, f(θ):

$$T_{rad} \cong f(\theta)T_c + \left(1 - f(\theta)\right)T_s \quad + \tag{A6}$$

With information about T$_{rad}$, $LAI$, and radiative forcing, the TSEB evaluates the soil (subscript 's') and canopy ('c') energy budgets separately, computing system and component fluxes of $R_n$, $H$, and $ET$ (i.e. $ET$ from soil and canopy), and $G$. Because angular effects are incorporated in Eq. A6, the TSEB can accommodate thermal data acquired at off-nadir viewing angles by geostationary satellites. The TSEB has a built-in mechanism for detecting thermal signatures of stress in the soil and canopy.





An initial iteration assumes that $T$ is occurring at potential (non-moisture limited) rate, while $E_{soil}$ is computed as a residual to the system energy budget. If the vegetation is stressed and transpiring at significantly less than the potential rate, $T$ will be overestimated and the residual $E_{soil}$ will become negative. Condensation onto the soil is unlikely midday on clear days, and therefore $E_{soil} < 0$ is considered a signature of system stress. Under such circumstances, $T$ is iteratively down-regulated until

$E_{soil} = 0$, noting that this assumption has been challenged by recent observations in some ecosystems (Perez-Priego et al., 2018).

For regional-scale applications of the TSEB, air temperature boundary conditions are difficult to specify with adequate accuracy due to localized land-atmosphere feedback. To overcome this limitation, the TSEB has been coupled with an atmospheric boundary layer (ABL) model, thereby simulating land-atmosphere feedback internally. In ALEXI (Anderson,

1997; Anderson et al., 2007), the TSEB is applied at two times during the morning ABL growth phase (between sunrise and local noon) using TIR from geostationary satellites. Energy closure over this interval is provided by a simple slab ABL model (McNaughton and Spriggs, 1986), which relates the rise in $T_a$ in the mixed layer to the time-integrated influx of H from the land surface. As a result, ALEXI uses only time-differential temperature signals, thereby minimizing flux errors due to absolute sensor calibration and atmospheric and emissivity corrections (Kustas et al., 2001). For local scale applications on length scales

similar to many flux footprints on the order of 100 m, the coarse-scale flux estimates can be spatially disaggregated using the DisALEXI technique (Norman et al., 2003). DisALEXI uses air temperature diagnosed by ALEXI at a nominal blending height along with high resolution LAI and LST data from polar orbiting satellites to estimate fluxes at finer scales.

**A2 Appendix B: ETRHEQ**

The ETRHEQ method is based on the hypothesis that gsurf at the daily time scale minimizes the vertical variance of RH in

the atmospheric boundary layer (Rigden and Salvucci, 2015; Salvucci and Gentine, 2013). The principles of ETRHEQ derive from complementary theory (Bouchet, 1962) as originally embedded in the PT-JPL remote sensing model (Fisher et al., 2008) and later used to update the PM-MOD16 model (Mu et al., 2011). ETRHEQ compares favorably to eddy covariance-measured $ET$ (Gentine et al., 2016; Rigden and Salvucci, 2016) and recent approaches have noted that it can be used to separate $E$ and $T$ (Rigden et al., 2018) by decomposing the contribution of $g_c$ and $g_{soil}$ to $g_{surf}$. Results from this partitioning approach compare

similarly to SIF estimates of modeled $T$ (Rigden et al., 2018), although both SIF measurements and the ETRHEQ partitioning algorithm could benefit from independent measures of $E$ and $T$.

**Author contributions**

PCS, TE-M, JBF, TG, OP-P, and THS performed analyses used to create figures and PCS performed analyses used to create Table 2. PCS, JBF, PG, TG, SPG, SL, DGM, and GW were responsible for writing subsections of the manuscript, and all

authors contributed to writing the text.



## Acknowledgements

PCS acknowledges support from the Alexander von Humboldt Foundation, the U.S. National Science Foundation award numbers 1552976, 1241881, and 1702029, the Montana Wheat and Barley Committee, and the USDA National Institute of Food and Agriculture Hatch project 228396 and Multi-State project W3188. Gabriel Bromley, Adam Cook, Elizabeth Vick, and Skylar Williams assisted with data collection and analysis, Chuck Merja, Dr. Scott Powell, Dr. James Irvine, and Dr. Bruce Maxwell provided logistical support, and Dr. Elke Eichelmann provided valuable comments to earlier drafts of this manuscript. DGM and WM acknowledge support from the European Research Council (ERC) under grant agreement 715254 (DRY–2– DRY) and the Belgian Science Policy Office (BELSPO) in the frame of the STEREO III programme project STR3S (SR/02/329). IM acknowledges the Academy of Finland Center of Excellence (projects No. 272041 and 118780) and ICOS-Finland (project No. 281255). TEM, MM, and OPP thank the Alexander von Humboldt foundation for financial support through the Max Planck Research Prize to MR. MM acknowledges the support of the Training on Remote Sensing for Ecosystem Modelling (Trustee) European Commission project no H2020-MSCA-ITN-2016-721995. THS and RGA acknowledge the USDA-ARS Office of National Programs (project number 2036-61000-018-00-D). JBF contributed to this research at the Jet Propulsion Laboratory, California Institute of Technology, under a contract with the National Aeronautics and Space Administration. California Institute of Technology. Government sponsorship acknowledge. JBF was supported in part by NASA programs: SUSMAP, WWAO, INCA, IDS, and ECOSTRESS. GW acknowledges financial support by the Austrian National Science Fund (FWF, grant no. P27176 and I3859). RP was supported by the Spanish MINECO-MICINN grant CGL2014-55583-JIN.

We also acknowledge the financial support to the eddy covariance data harmonization provided by CarboEuropeIP, FAO-GTOS-TCO, iLEAPS, Max Planck Institute for Biogeochemistry, National Science Foundation, University of Tuscia, Université Laval, Environment Canada and US Department of Energy and the database development and technical support from Berkeley Water Center, Lawrence Berkeley National Laboratory, Microsoft Research eScience, Oak Ridge National Laboratory, University of California – Berkeley and the University of Virginia. We would additionally like to thank the North American Carbon Program Site-Level Interim Synthesis team, the Modeling and Synthesis Thematic Data Center, and the Oak Ridge National Laboratory Distributed Active Archive Center, and Larry Flanagan for collecting, organizing, and distributing the model output used in this analysis. The authors want to thank Silvana Schott for her support in producing Figure 7.

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



## Tables

**Table 1: A summary of recent approaches for estimating transpiration (T) and/or for partitioning evapotranspiration (ET) into evaporation (E) and T at the ecosystem scale. The reader is referred to Kool et al. (2014) for a comprehensive review E and T measurement methodologies.**

| Approach | Advantages | Disadvantages | Selected References |
|---|---|---|---|
| Flux variance similarity | Uses high-frequency eddy covariance data. Open source software is available | Necessary terms rarely computed and/or high frequency data to calculate terms rarely shared. Sensitive to water use efficiency assumptions. | Scanlon and Kustas (2010); Scanlon and Sahu (2008); Skaggs et al. (2018) |
| Analyses of half-hourly to hourly eddy covariance data | Use widely-available eddy covariance data | Often rely on assumptions regarding water use efficiency and the maximum value of the T/ET ratio | Berkelhammer et al. (2016); Lin et al. (2018); Li et al. (2019); Scott and Biederman (2017); Zhou et al. (2016) |
| Solar-induced fluorescence | Measurements are available at ecosystem to global scales. | Relies on an empirical relationship between T and gross primary productivity; mechanistic link not yet understood. Uncertainty in SIF retrieval. | Damm et al. (2018; Lu et al. (2018; Shan et al. (2019) |
| Carbonyl sulfide (COS) flux | Can be measured using eddy covariance techniques to estimate canopy conductance. | COS flux can also arise from non-stomatal sources. | Whelan et al. (2018; Wohlfahrt et al. (2012) |





**Table 2: The exponential term (m) of the model EWUE = kD^m fit using nonlinear least squares to the 95th percentile of EWUE values in 0.3 kPa bins of D (see Fig. 4). Sites: CA-Ca1 (Schwalm et al., 2007), CA-Obs (Griffis et al., 2003; Jarvis et al., 1997), US-Ho1 (Hollinger et al., 1999). Models: BEPS (Liu et al., 1999), CAN-IBIS (Williamson, 2008), CNCLASS (Arain et al., 2006), ECOSYS (Grant et al., 2005), ED2 (Medvigy et al., 2009), ISAM , ISOLSM (Riley et al., 2002), LOTEC (Hanson et al., 2004), ORCHIDEE (Krinner et al., 2005), SIB (Baker et al., 2008), SIBCASA (Schaefer et al., 2009), SSIB2 (Zhan et al., 2003), TECO (Weng and Luo, 2008). Data are available from Ricciuto et al. (2013).**

| Model | CACa1 | CAObs | USHo1 |
|---|---|---|---|
| BEPS | -0.6 | -0.5 | -0.5 |
| CAN-IBIS | -4.8 | -2.5 | -4.7 |
| CNCLASS | -3.3 | -4.1 | -3.3 |
| ECOSYS | -2.3 | -1.6 | -0.7 |
| ED2 | -2.1 | -2.4 | -3.4 |
| ISAM | -4.8 | -0.9 | -1.3 |
| ISOLSM | -4.7 | -0.8 | -1.4 |
| LOTEC | -4.7 | -2.2 | -4.4 |
| ORCHIDEE | -2.3 | -3.9 | -4.5 |
| SIB | -3.6 | -2.8 | -2.3 |
| SIBCASA | -4.5 | -2.9 | -3.6 |
| SSIB2 | -4.7 | -3.1 | -4.1 |
| TECO | -4.1 | -3.3 | -1.8 |
| **Measurements** | **-0.5** | **-0.4** | **-0.5** |




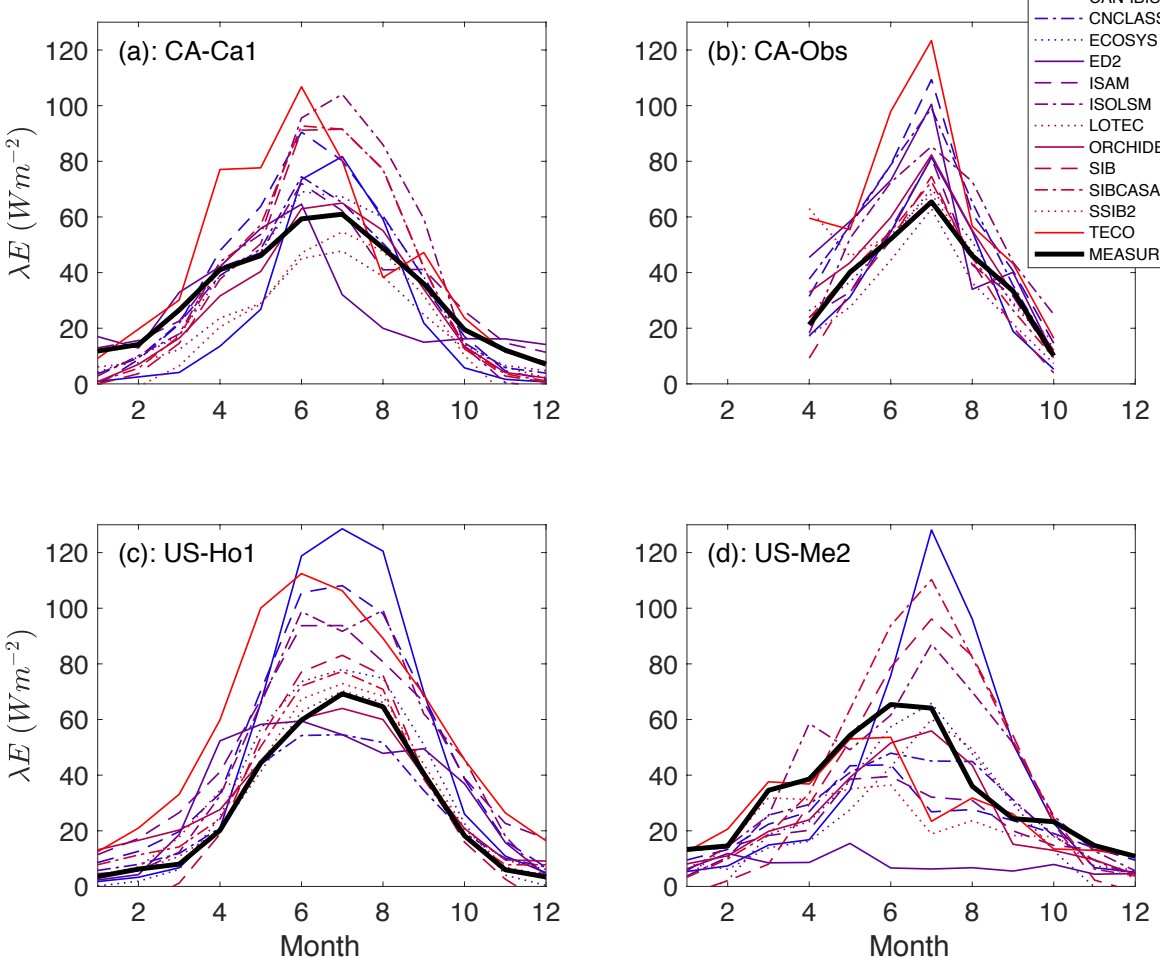

**Figure 1: The mean monthly latent heat flux ($\lambda E$) - the energy used for evapotranspiration - from eddy covariance measurements from four research sites ('MEASURED') and 13 ecosystem models from the North American Carbon Program Site-Level Interim Synthesis (Schwalm et al., 2010). Sites: CA-Ca1 (Schwalm et al., 2007), CA-Obs (Griffis et al., 2003; Jarvis et al., 1997), US-Ho1 (Hollinger et al., 1999), US-Me2 (Thomas et al., 2009). Models: BEPS (Liu et al., 1999), CAN-IBIS (Williamson, 2008), CNCLASS (Arain et al., 2006), ECOSYS (Grant et al., 2005), ED2 (Medvigy et al., 2009), ISAM (Jain and Yang, 2005), ISOLSM (Riley et al., 2002), LOTEC (Hanson et al., 2004), ORCHIDEE (Krinner et al., 2005), SIB (Baker et al., 2008), SIBCASA (Schaefer et al., 2009), SSIB2 (Zhan et al., 2003), TECO (Weng and Luo, 2008). Data are available from Ricciuto et al. (2013).**





**Figure 2: The relationship between above-canopy vapor pressure deficit (_D_) and evapotranspiration (_ET_) visualized using Kernel Density Estimation (Botev et al., 2010) for more than 1.5 million half hourly eddy covariance observations with a solar zenith angle less than 60° from 241 eddy covariance research sites in the La Thuile FLUXNET database that included ecosystem type and soil**
5   **heat flux measurements described in Stoy et al. (2013).**





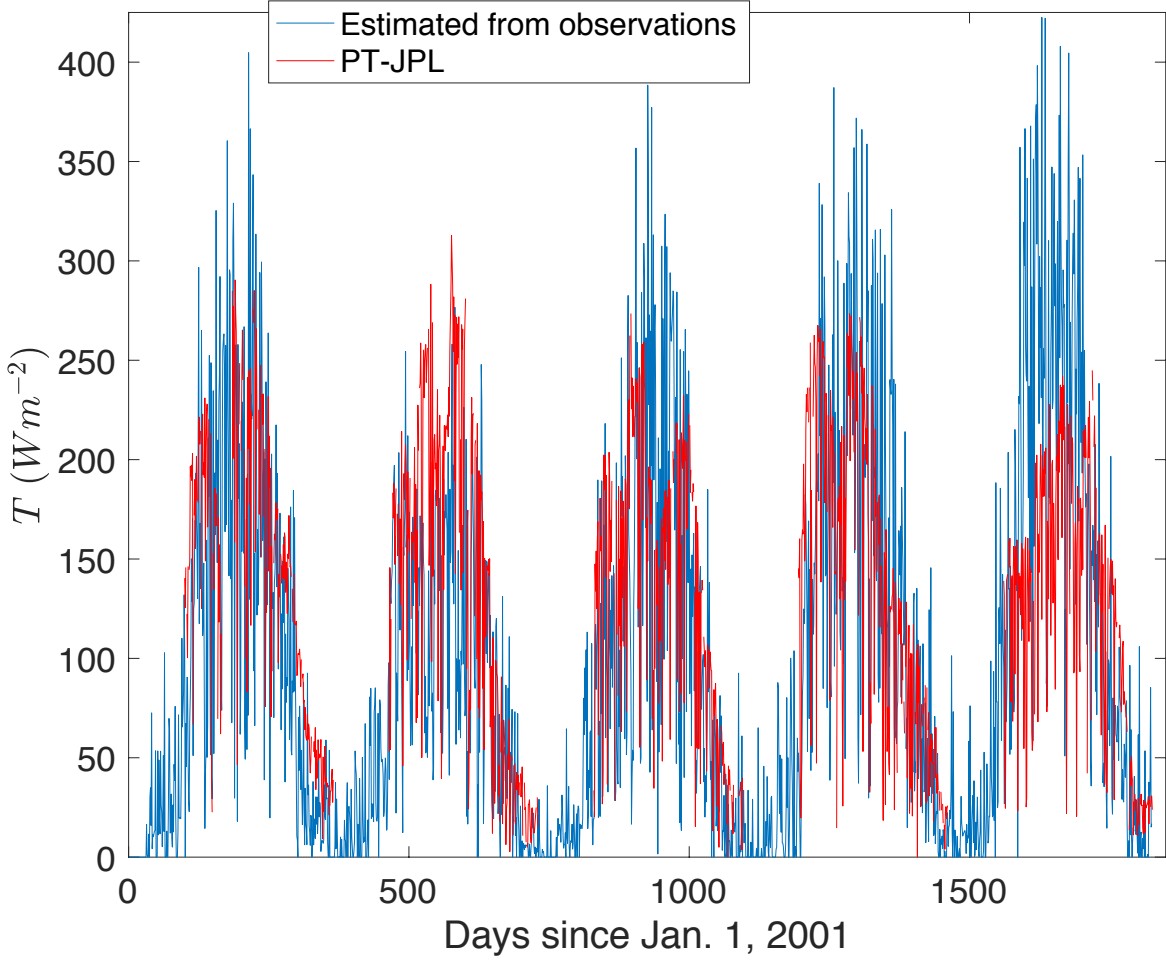

**Figure 3: The Priestley - Taylor Jet Propulsion Lab (PT-JPL) estimate of transpiration (*T*) in energy flux units compared against *T* estimated using eddy covariance measurements and models of soil evaporation in a loblolly pine forest for 2001-2005 from Stoy et al. (2006). Measurements were taken at 10:30 Eastern Standard Time (UTC - 5:00).**





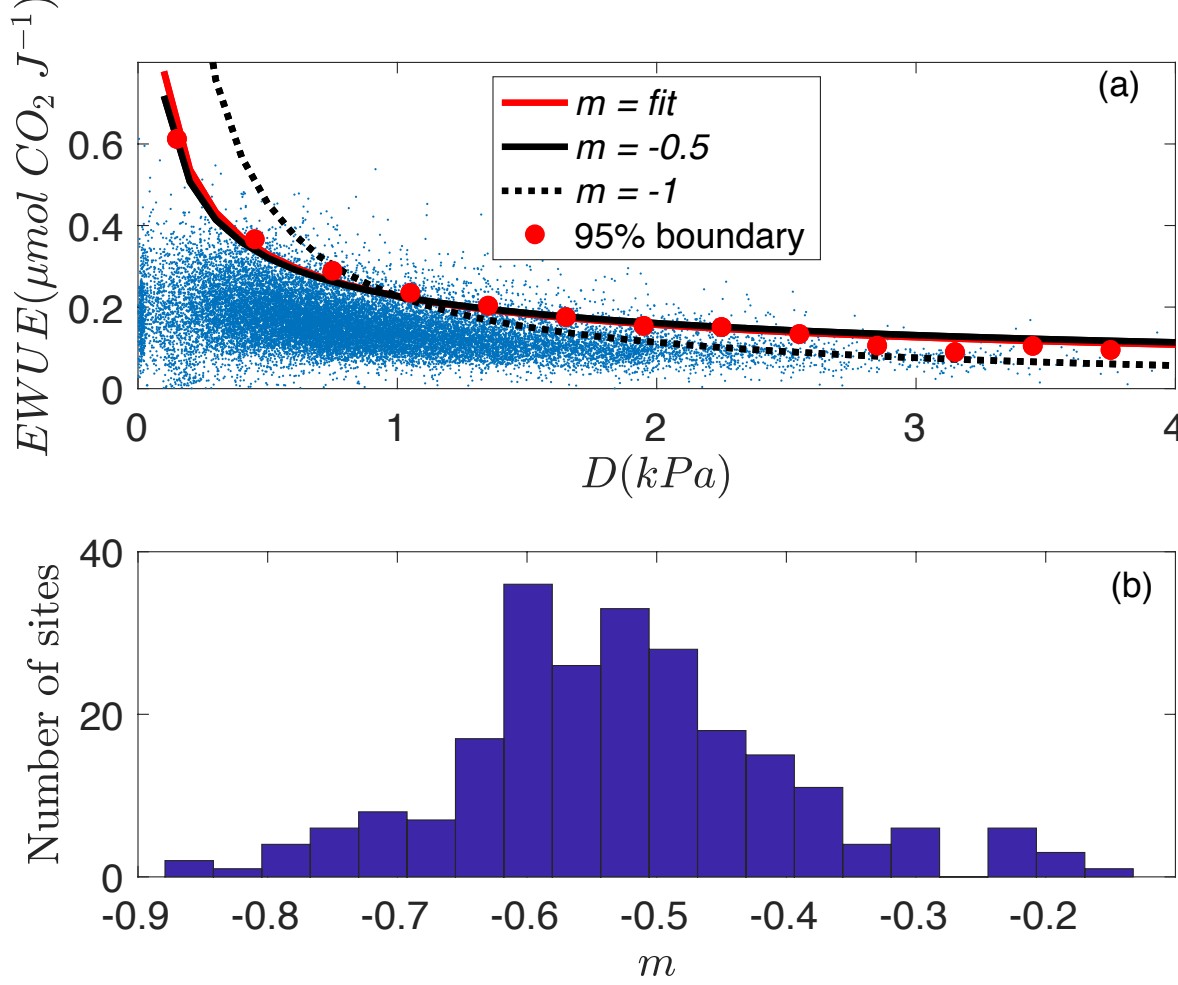

**Figure 4: A. An example of a boundary-line analysis for the relationship between vapor pressure deficit (*D*) and ecosystem water use efficiency (*EWUE*) for the case of a single ecosystem in the LaThuile FLUXNET database, in this case Vielsalm, Belgium (BE-Vie) using *EWUE* values that represent the 95th percentile of 0.3 kPa D bins and parameters of the model *EWUE* = k*D*<sup>m</sup> fit using**

5  **nonlinear least squares. The value of *m* for BE-Vie is -0.53; values *m* = -0.5 following (Medlyn et al., 2011) and *m* = -1 following (Leuning, 1995) are shown for reference with the same fitted value of k. B. The distribution of the best-fit exponential parameter (m) for 240 sites in the La Thuile FLUXNET database that contained full energy balance measurements and ecosystem type information used in Stoy et al. (2013).**





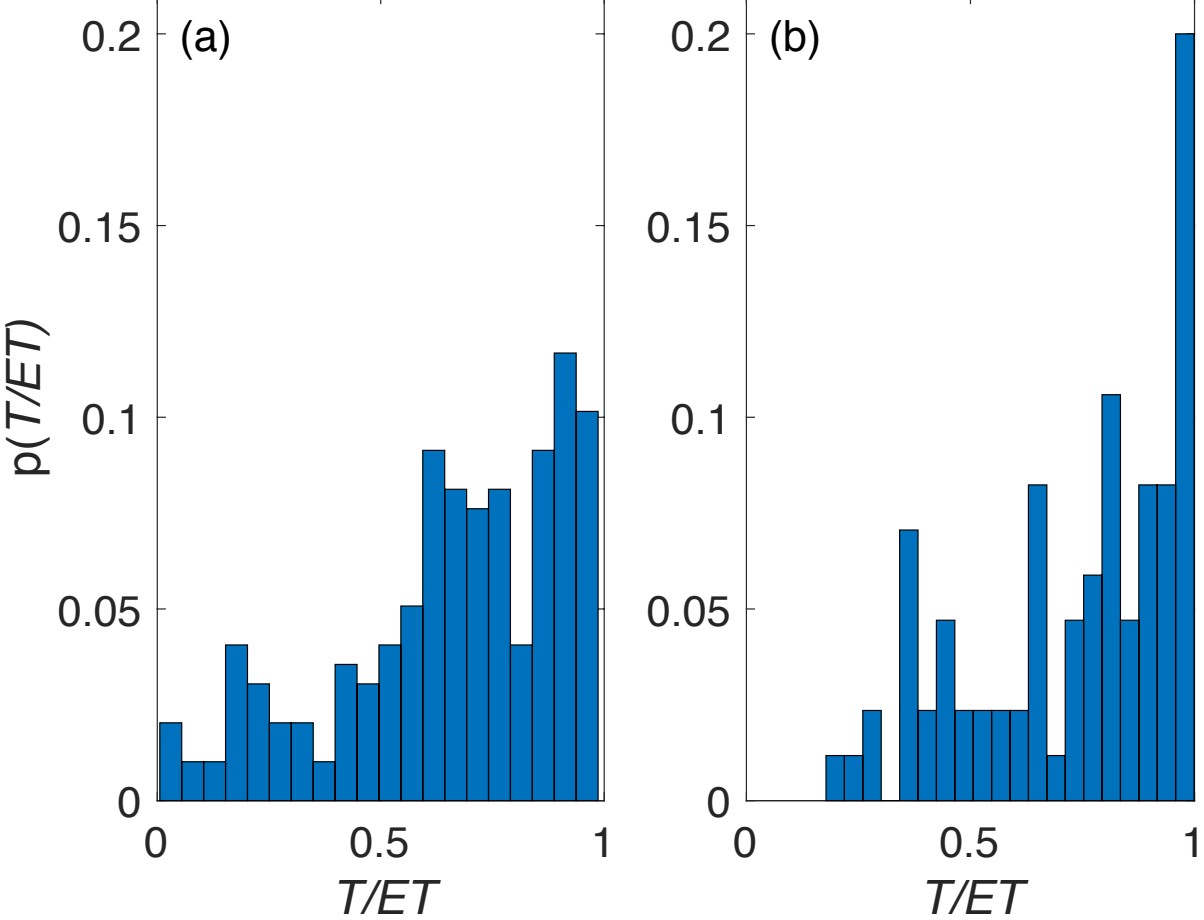

**Figure 5: The distribution of fraction of evapotranspiration arising during daytime hours from transpiration (*T*) and evaporation (*E*) estimated using the flux variance similarity approach of Scanlon and Kustas (2010) from (a) a winter wheat field near Moore, MT, USA described in Vick et al. (2016) using a version of the original algorithm, and (b) a winter wheat field near Sun River, MT using Fluxpart (Skaggs et al., 2018).**





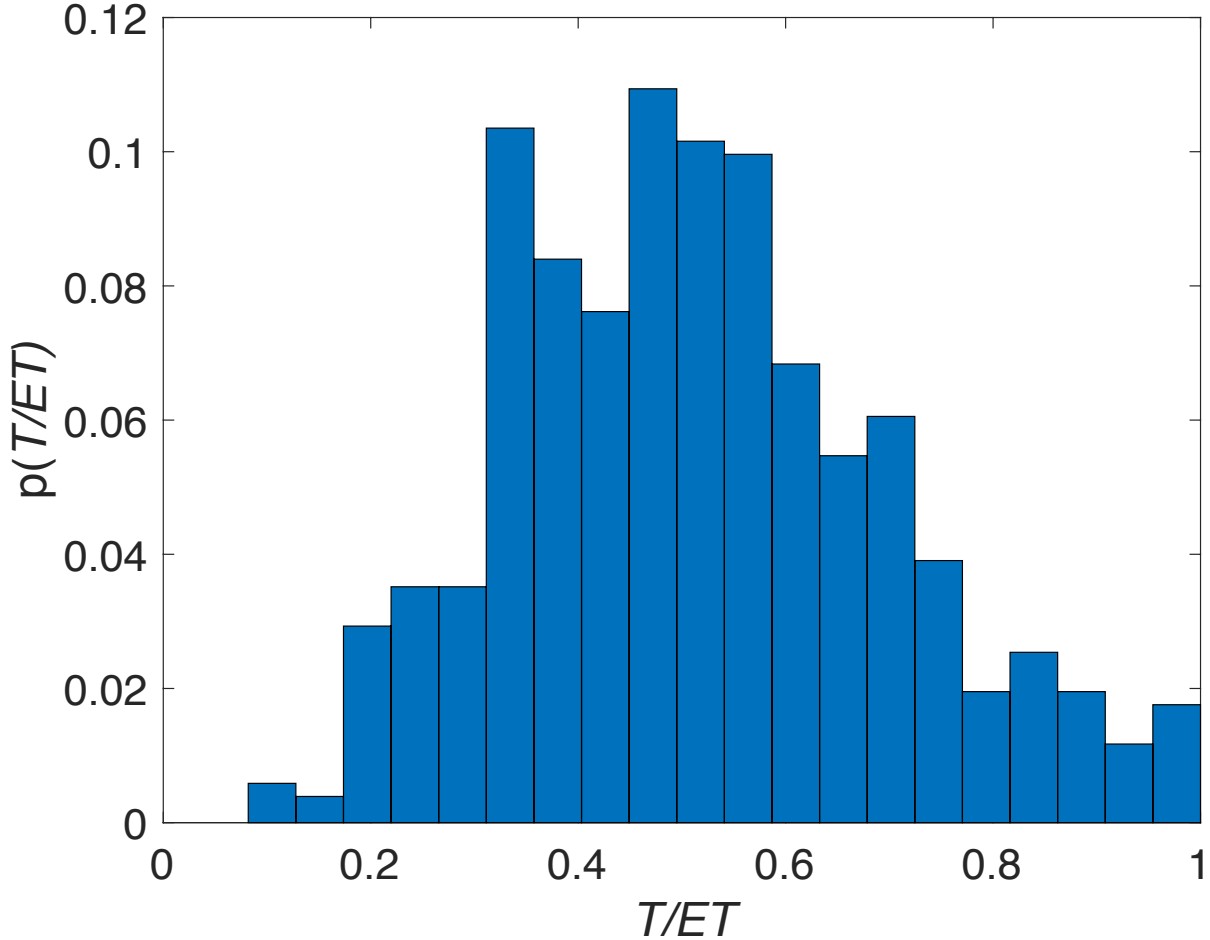

**Figure 6: The distribution of the *T/ET* ratio for half hourly observations from the partitioning approach of Perez-Priego et al. (2018) for the Majadas de Tietar (ES-Lma), Spain research site.**





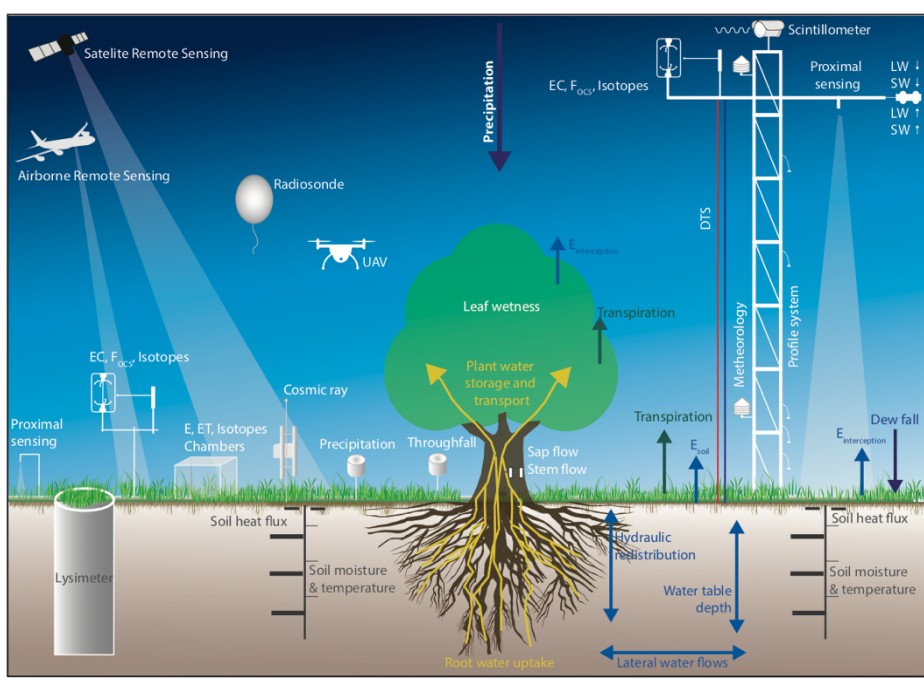

**Figure 7: A schematic of an ecosystem experiment designed to measure transpiration and evaporation from soil and intercepted water using multiple complementary measurement approaches.**