# Peer review of "Reviews and syntheses: Turning the challenges of partitioning ecosystem evaporation and transpiration into opportunities"

_Biogeosciences, 2019_

## Referee Comment (RC1) · Anonymous Referee #1 · 17 Apr 2019

General comments:

The manuscript reviews Evapotranspiration partitioning methods, with focus on the most recent ones; and because of the links to photosynthesis and physiology it is well placed in BG. Beyond a pure review, interesting and novel own considerations of the authors are added which might help to test and improve these methods in the future.

It is well written, and gives an impressing complete and detailed overview on those aspects of ET partitioning the authors chose to treat in-depth. Concerning this choice, I have one general comment on the labelling/scope/structure of the manuscript. The title and large parts of abstract, introduction and background seem to suggest that the

whole field of ET partitioning is subject to the paper. Later during the manuscript it becomes clear that three categories of partitioning approaches are treated differently profound.

First, in section 3.0 a number of methods are mentioned to be outside the scope of the manuscript (because of multiple recent other reviews), but nevertheless most of them are more or less briefly mentioned in the following paragraph, which is a bit confusing. Maybe it would help clarity to either not write anything about them that goes beyond a mere list, or to treat them a bit more detailed (e.g. with one reference per method) but then adapt the way they are placed in context of the paper (i.e. exchange "not our intent to reiterate them" for something like "will only give a brief overview", or put this statement after the list of methods rather than before it). Also, the role of bulk ET methods such as watershed residual or scintillometry, while surely worth mentioning somewhere in the MS, is not clear at its particular place between the above statement and the partitioning methods. If it stays here, it should be better linked to the text around. Finally, I wonder whether subcanopy EC measurements (now in 3.1 at p6L6) might better fit in this section too. Technically, they are indeed half-hourly EC observations once they have been installed, but this is the case at few stations, done deliberately for partitioning, they have a small footprint, and no further connection to what is discussed in 3.1 (or the rest of the MS before Sect. 5). In all this, they resemble the methods in 3.0. I am not sure whether "scale" is the ideal criterion to distinguish 3.0 from the other methods, but I have no better suggestion either.

Second, there are methods in section 3.2 to 3.7 which are treated as in a typical review – summarizing the latest state of the art very well as far as I can judge; maybe a bit more detail and explanation would help at some points.

Third, the methods in section 3.1 appear to be at the heart of the MS. They are not only summarized very thoroughly, but section 4.1 and 4.2 also present considerations that, while of general interest, are particularly valuable for the assessment and improvement of these methods (and to some degree of those in 3.2).

I would like to encourage the authors, if not amending the MS such that all methods receive similar attention (which would be a major revision and surely is an option, but probably not the intention and maybe not so interesting given the existing reviews), to think where minor revisions to the wording of title, abstract, intro and background can help give readers a clearer impression of the focus of the paper.

Specific comments:

p6L20: reword or explain in more depth the deltas and the word marginal in this specific context

p6L25: Maybe VPD instead of D would help readers easily recognize the variable all over all over the manuscript?

p8L4: The title "Advanced algorithms for partitioning eddy covariance data" is somewhat arbitrary as a distinction from the section before. Maybe something like "Partitioning ET using high-frequency Eddy-covariance raw data"?

p8L25: The better results during fair weather were not a result of the LES comparison. One more maybe noteworthy result of the LES comparison was, however (if it is not too detailed for the intention of this review) that an assumption about transfer efficiencies in the original Scanlon approach is frequently violated.

p9L16: The various definitions of ecosystem- (as opposed to leaf-) level WUE seem to become more and more confusing. After WUEeco = NEE/ET (Scanlon and Sahu 2008) and WUEeco = GPP/ET (Beer et al. 2009, Global Biogeochem. Cycles 23: GB2018), this one is already the 3rd. While this is not the fault of the review authors, they might want to take the opportunity to try to order them a bit or at least mention the variety of existing definitions. The three above, in that order, can be thought of as increasingly close approximations of leaf-level WUE. While all have their methodological justifications, it seems counter-intuitive to label the one closest to leaf-WUE (i.e. the 3rd) as "ecosystem-level". IMHO a reader coming across that term for the first time,

would rather expect it to indicate the ratio of CO2 gained (in net) by the ecosystem to vapour spent by the (whole) ecosystem, i.e. the first one (NEE/ET).

p11L1-13: The first paragraph summarizes how satellite-based remote sensing can be used to quantify bulk ET (without mentioning partitioning) while the second one on partitioning seems to apply only to much lower/closer remote sensing platforms (tower or airborne). If this is true, please try to put more clearly. Otherwise (e.g. if the separate E and T estimates occurring in some satellite-based ET algorithms have been proposed as serious partitioning method, rather than just means of minimizing the bulk ET error), add such information.

p13L12: Consider comment p9L16 as to how to call this type of WUE.

p13L16-19: The methodological details of this interesting approach and their effects on interpretation could be elaborated a bit more. Was T for the left-hand side of the equation / the Y-axis of the figure explicitly needed? My guess from the text (but this is not completely clear) is that you used GPP/ET from the Fluxnet dataset, and the task (or at least one of the tasks) of the 95-percentile separation was to extract the data points where T->ET. If, in contrast, T was explicitly determined from the EC data, which method was used to avoid any circular reasoning? Also, it would be interesting to learn whether the 95-percentile rather ruled out specific ecosystems (that fail to behave "optimally"), specific meteorological situations, or a mixture of both. Is the result sensitive towards changing the percentile (e.g. 90 or 99 %)?

Fig2: Are the restrictions mentioned in the caption (solar zenith angle, soil heat flux, ecosystem info) motivated by this study? Otherwise it may be sufficient to mention the dataset from Stoy et al. 2013 was used, or to mention somewhere (caption or discussion text) the only one(s) that might possibly have had an unwanted systematic side effect on the relation suggested by the figure (which is probably solar zenith angle).

Technical corrections:

p10L17: T in italics

p12L14: though => through?

p15L19: gages => gauges?

Fig. 2: Colour-coded values are probability densities with unit 1/([ET]*[D])? Is there one h too much in the unit of ET on the Y axis?

---

## Short Comment (SC1) · 22 Apr 2019

Thank you for this nice review giving further insights into new partitioning approaches for evapotranspiration.

As far as I can tell, a typo crept into page 8, line 13 and line 19: Scanlon and Sahu (2008) introduced for the variance of $CO_2$ due to stomatal uptake the variable $\sigma_{cp}$, and "cr", however, stood for the soil $CO_2$ source. Thus, maybe to avoid confusion with the former publications about this method, this variable label could be adapted.

---

## Referee Comment (RC2) · Zhongwang Wei (Referee) · 4 May 2019

In this study, the authors reviewed current progress in partitioning soil/canopy interception evaporation and canopy transpiration. The review is impressive and I suspect that it will stand out among previous studies. In addition, they also provided a perspective on how to improve the involved theory and observations. it is well written, organized, and easy to understand. I think a minor revision is required before it can be considered for a publication in Biogeosciences.

Major Comments:

[Figure]

1. The manuscript structure is a little bit confusing. In my opinion, the title of Sec. 3.2 should be changed in order to distinguish from 3.1. Or simply merge 3.1 and 3.2 together. It is better to move Sec. 3.6 to other section.

2. It is better to briefly introduce the method performance at different time scales. For example, a stable state isotopic assumption may work well in daily time scale but fail in an hourly or sub-hourly time scale. While Zhou et al. (2015) and (2016) found the underlying water use efficiency method works well at the half-hourly and daily time scales both.

3. Additional review of a novel and direct method for ET partitioning method proposed by Or and Lehmann (2019) is suggested. This method is useful and Unique. This will make the paper more complete.

Or, D., & Lehmann, P. (2019). Surface evaporative capacitance: How soil type and rainfall characteristics affect global- scale surface evaporation. Water Resources Research, 55. https://doi.org/10.1029/2018WR024050

Minor comments:

1. P2L2 some ecosystems, but other ecosystems do: please specify.

2. P2L28 I suggest adding the method proposed by Or, D., & Lehmann, P. (2019) in Table 1.

3. P3L3 The ratio of transpiration to evapotranspiration: long term?

4. P4L7 Rn-G is another uncertainty source, especially for the wetland.

5. P7L12 (Perez-Priego et al., 2018): Perez-Priego et al. (2018)

6. P9L24 Alemohammad et al., 2017; Damm et al., 2018; Lu et al., 2018; Pagán et al., 2019; Shan et al., 2019): (Alemohammad et al., 2017; Damm et al., 2018; Lu et al., 2018; Pagán et al., 2019; Shan et al., 2019)

7. P9L32 FCOS (pmol m-2 s-1): FCOS (pmol mËE̞{-2} s{-1})

8. P14 Sec.4.2 This question is great. For some vegetation types, T/ET is easy to reach unity not only because of high canopy conductance but also e.g. tree morphological, soil/root condition and human impact. For example, crops have high $T/(E + T)$ under low LAI conditions potentially influenced by human effects (a high water use efficiency and less constrained by environmental stresses). Please check Wei et al. (2017) for further information.

9. P17L25: Please also introduce the measurement of radiometric surface temperature.

10. P18L19 gsurf: $g_{surf}$

11. P16L14 (Fisher et al., 2008): Fisher et al. (2008)

12. P16L15 (Jarvis, 1976) : Jarvis (1976)

13. P17L24 Norman et al. )1995): Norman et al. (1995)
* * *

---

## Author Comment (AC1) · 24 May 2019

Thank you for your interest in the manuscript and for noticing this error, which is now corrected in the revised manuscript, and thank you for the other contributions to the text that you provided.

---

## Author Comment (AC2) · 24 May 2019

General comments:

The manuscript reviews Evapotranspiration partitioning methods, with focus on the most recent ones; and because of the links to photosynthesis and physiology it is well placed in BG. Beyond a pure review, interesting and novel own considerations of the authors are added which might help to test and improve these methods in the future.

*Thank you for your insightful comments and support of the manuscript.*

It is well written, and gives an impressing complete and detailed overview on those aspects of ET partitioning the authors chose to treat in-depth. Concerning this choice, I have one general comment on the labelling/scope/structure of the manuscript. The title and large parts of abstract, introduction and background seem to suggest that the whole field of ET partitioning is subject to the paper. Later during the manuscript it becomes clear that three categories of partitioning approaches are treated differently.

First, in section 3.0 a number of methods are mentioned to be outside the scope of the manuscript (because of multiple recent other reviews), but nevertheless most of them are more or less briefly mentioned in the following paragraph, which is a bit confusing. Maybe it would help clarity to either not write anything about them that goes beyond a mere list, or to treat them a bit more detailed (e.g. with one reference per method) but then adapt the way they are placed in context of the paper (i.e. exchange "not our intent to reiterate them" for something like "will only give a brief overview", or put this statement after the list of methods rather than before it). Also, the role of bulk ET methods such as watershed residual or scintillometry, while surely worth mentioning somewhere in the MS, is not clear at its particular place between the above statement and the partitioning methods. If it stays here, it should be better linked to the text around. Finally, I wonder whether subcanopy EC measurements (now in 3.1 at p6L6) might better fit in this section too. Technically, they are indeed half-hourly EC observations once they have been installed, but this is the case at few stations, done deliberately for partitioning, they have a small footprint, and no further connection to what is discussed in 3.1 (or the rest of the MS before Sect. 5). In all this, they resemble the methods in 3.0. I am not sure whether "scale" is the ideal criterion to distinguish 3.0 from the other methods, but I have no better suggestion either.

*We did not want to exclude any method while maintaining a focus on whole-ecosystem evaporation and transpiration partitioning approaches. We re-organized the beginning of section 3 in response to this comment and the comments of Reviewer 2.*

Second, there are methods in section 3.2 to 3.7 which are treated as in a typical review – summarizing the latest state of the art very well as far as I can judge; maybe a bit more detail and explanation would help at some points.

*We sought to write a succinct overview of each method and elicited the help of additional contributors to ensure that each received an optimal amount of attention.*

Third, the methods in section 3.1 appear to be at the heart of the MS. They are not only summarized very thoroughly, but section 4.1 and 4.2 also present considerations that, while of

general interest, are particularly valuable for the assessment and improvement of these methods (and to some degree of those in 3.2).

I would like to encourage the authors, if not amending the MS such that all methods receive similar attention (which would be a major revision and surely is an option, but probably not the intention and maybe not so interesting given the existing reviews), to think where minor revisions to the wording of title, abstract, intro and background can help give readers a clearer impression of the focus of the paper.

*We hope that our focus on whole-ecosystem evaporation and transpiration partitioning approaches did not overemphasize the importance of one measurement technique over any other and hope that our restructuring further improves balance across the manuscript. We critiqued and made improvements to all sections of the manuscript in response to these comments.*

Specific comments:
p6L20: reword or explain in more depth the deltas and the word marginal in this specific context

*We now define delta in the equation, thank you for suggesting this. 'Marginal' water use efficiency in this case is the change in transpiration per unit change in evaporation. This notion arises from Cowan and Farquhar (1977) and subsequent references and happens to be described nicely as Lagrangian multiplier in Manzoni et al. (2011) and similar references, but rather than a lengthy discussion of marginal gains and optimality theory in the present manuscript we decided to keep a simpler description in the interest of space.*

p6L25: Maybe VPD instead of D would help readers easily recognize the variable all over all over the manuscript

*We discussed the use of VPD, but D is also common and shorter and we decided to keep this abbreviation.*

p8L4: The title "Advanced algorithms for partitioning eddy covariance data" is some- what arbitrary as a distinction from the section before. Maybe something like "Partition- ing ET using high-frequency Eddy-covariance raw data"?

*Thank you for the suggestion, we changed the subsection header per your recommendation.*

p8L25: The better results during fair weather were not a result of the LES comparison. One more maybe noteworthy result of the LES comparison was, however (if it is not too detailed for the intention of this review) that an assumption about transfer efficiencies in the original Scanlon approach is frequently violated.

*We removed the statement regarding fair weather which applies to all eddy covariance measurements that are usually not possible during rain. We also requested assistance from an expert, Dr. Anne Klosterhalfen, who helped us improve this section.*

p9L16: The various definitions of ecosystem- (as opposed to leaf-) level WUE seem to become more and more confusing. After WUEeco = NEE/ET (Scanlon and Sahu 2008) and WUEeco = GPP/ET (Beer et al. 2009, Global Biogeochem. Cycles 23: GB2018), this one is already the 3rd. While this is not the fault of the review authors, they might want to take the opportunity to try to order them a bit or at least mention the variety of existing definitions. The three above, in that order, can be thought of as increasingly close approximations of leaf-level WUE. While all have their methodological justifications, it seems counter-intuitive to label the one closest to leaf-WUE (i.e. the 3rd) as "ecosystem-level". IMHO a reader coming across that term for the first time, would rather expect it to indicate the ratio of $CO_2$ gained (in net) by the ecosystem to vapour spent by the (whole) ecosystem, i.e. the first one (NEE/ET).

*This is an interesting point and we agree that the multiple definitions of water use efficiency make things unnecessarily complicated. We also agree that NEE/ET is probably better defined as 'ecosystem water use efficiency' and from this perspective the figure y-axis was a bit mis-leading, not by intent. We revisited each instance of 'WUE' or 'EWUE' throughout the manuscript and re-worded text when necessary to be entirely clear in each case.*

p11L1-13: The first paragraph summarizes how satellite-based remote sensing can be used to quantify bulk ET (without mentioning partitioning) while the second one on partitioning seems to apply only to much lower/closer remote sensing platforms (tower or airborne). If this is true, please try to put more clearly. Otherwise (e.g. if the separate E and T estimates occurring in some satellite-based ET algorithms have been proposed as serious partitioning method, rather than just means of minimizing the bulk ET error), add such information.

*This is an important point. We want to note first how remote sensing platforms can estimate ET in principle as a first step for explaining how high-resolution remote sensing (e.g. from tower mounted cameras) can use the same principles to actually measure T directly. We edited the text to make this point clearer.*

p13L12: Consider comment p9L16 as to how to call this type of WUE.

*Per the comments above and comments of Reviewer 2, we changed our description of different WUE terms to be extremely explicit throughout the manuscript.*

p13L16-19: The methodological details of this interesting approach and their effects on interpretation could be elaborated a bit more. Was T for the left-hand side of the equation / the Y-axis of the figure explicitly needed? My guess from the text (but this is not completely clear) is that you used GPP/ET from the Fluxnet dataset, and the task (or at least one of the tasks) of the 95-percentile separation was to extract the data points where T->ET. If, in contrast, T was explicitly determined from the EC data, which method was used to avoid any circular reasoning? Also, it would be interesting to learn whether the 95-percentile rather ruled out specific ecosystems (that fail to behave "optimally"), specific meteorological situations, or a mixture of both. Is the result sensitive towards changing the percentile (e.g. 90 or 99 %)?

*Yes, your interpretation is correct. Our approach does assume that ET approaches T in some instances, and that the 'edge' of the relationship between D and EWUE defined as GPP/T can be*

*approximated by GPP/ET when ET approaches T. Optimality theory predicts that this quantity will be constrained by D. We simply use the 95% threshold as a value at which one might consider E to be trivial compared to T in the ET measurement to identify this constraint, and re-worded the text to explain our approach in more detail. The approach will be sensitive to changing the percentile, changing the bin size, and changing the cost function used to compute the 'm' parameter. Rather than explore these variables comprehensively in a sensitivity analysis, we simply note that observations are broadly consistent with the notion that ecosystem carbon and water fluxes are constrained by an optimal response to vapor pressure deficit. We do feel that an expanded analysis – apart from a review and prospectus manuscript – on this topic would be forthcoming.*

Fig2: Are the restrictions mentioned in the caption (solar zenith angle, soil heat flux, ecosystem info) motivated by this study? Otherwise it may be sufficient to mention the dataset from Stoy et al. 2013 was used, or to mention somewhere (caption or discussion text) the only one(s) that might possibly have had an unwanted systematic side effect on the relation suggested by the figure (which is probably solar zenith angle).

*We did select solar zenith angles that satisfied the conditions specified in the legend although yes, we did only use eddy covariance measurements that included soil heat flux values as in Stoy et al. (2013). For these reasons we felt that a succinct yet detailed description was important.*

Technical corrections:
p10L17: T in italics
p12L14: though => through?
p15L19: gages => gauges?

*T has now been italicized, 'through' is no longer misspelled, and gauges is the correct spelling, thank you for noticing these errors.*

Fig. 2: Colour-coded values are probability densities with unit 1/([ET]*[D])? Is there one h too much in the unit of ET on the Y axis?

*The units are mm per the half-hourly averaging interval of the eddy covariance measurements. We now define hh in the figure legend for clarity.*

**References**
Cowan, I.R. and Farquhar, G. D.: Stomatal function in relation to leaf metabolism and environment.

---

## Author Comment (AC3) · 24 May 2019

In this study, the authors reviewed current progress in partitioning soil/canopy interception evaporation and canopy transpiration. The review is impressive and I suspect that it will stand out among previous studies. In addition, they also provided a perspective on how to improve the involved theory and observations. it is well written, organized, and easy to understand. I think a minor revision is required before it can be considered for a publication in Biogeosciences.

*We thank Dr. Wei for insightful comments and we hope that we have addressed them all adequately.*

Major Comments:
1. The manuscript structure is a little bit confusing. In my opinion, the title of Sec. 3.2 should be changed in order to distinguish from 3.1. Or simply merge 3.1 and 3.2 together. It is better to move Sec. 3.6 to other section.

*Reviewer 1 also recommended a re-structuring of Section 3. We wanted to keep sections 3.1 (on half-hourly eddy covariance data) and sections 3.2 (on high frequency eddy covariance data) separate to emphasize that these use different data sources and assumptions. We like how section 3.6 serves as a bridge between remote sensing and isotopic approaches, but undertook a comprehensive re-structuring of section 3 following the insightful recommendation (below) to add the approach of Or and Lehmann (2019) and following the recommendations of Reviewer 1.*

2. It is better to briefly introduce the method performance at different time scales. For example, a stable state isotopic assumption may work well in daily time scale but fail in an hourly or sub-hourly time scale. While Zhou et al. (2015) and (2016) found the underlying water use efficiency method works well at the half-hourly and daily time scales both.

*All approaches have a range of time scales at which they are more applicable, and approaches that work at shorter time scales should in principle be able to scale up to larger time scales. We added text to explain the time scale sensitivity of different approaches while attempting to avoid an exhaustive discussion of the time sensitivity of each.*

3. Additional review of a novel and direct method for ET partitioning method proposed by Or and Lehmann (2019) is suggested. This method is useful and Unique. This will make the paper more complete.
Or, D., & Lehmann, P. (2019). Surface evaporative capacitance: How soil type and rainfall characteristics affect global- scale surface evaporation. Water Resources Re- search, 55. https://doi.org/10.1029/2018WR024050

*Thank you for suggesting the manuscript of Or and Lehmann, which was published shortly before our manuscript was submitted and likely would have escaped our notice. We added a subsection in section 3 that describes the fundamentals of this approach.*

Minor comments:
1. P2L2 some ecosystems, but other ecosystems do: please specify.

*We re-worded this passage for clarity.*

2. P2L28 I suggest adding the method proposed by Or, D., & Lehmann, P. (2019) in Table 1.

*We have added the Or and Lehman method to Table 1, thank you for the suggestion.*

3. P3L3 The ratio of transpiration to evapotranspiration: long term?

*T/ET in this instance refers to annual time scales (see Fig. 3 in Good et al., 2015) and we added this detail to the text.*

4. P4L7 Rn-G is another uncertainty source, especially for the wetland.

*This is correct, advective energy flux via moving water is a major challenge for closing the energy balance of wetlands. We added text to incorporate this notion.*

5. P7L12 (Perez-Priego et al., 2018): Perez-Priego et al. (2018)
6. P9L24 Alemohammad et al., 2017; Damm et al., 2018; Lu et al., 2018; Pagán et al., 2019; Shan et al., 2019): (Alemohammad et al., 2017; Damm et al., 2018; Lu et al., 2018; Pagán et al., 2019; Shan et al., 2019)

*Thank you for noting the referencing errors, which are now fixed.*

7. P9L32 FCOS (pmol m-2 s-1): FCOS (pmol mËE $_{\dot{c}}${-2} s{-1})

*This comment refers to incorrect superscripting of the $F_{COS}$ units. We have fixed them and thank you for noticing.*

8. P14 Sec.4.2 This question is great. For some vegetation types, T/ET is easy to reach unity not only because of high canopy conductance but also e.g. tree morphological, soil/root condition and human impact. For example, crops have high $T/(E + T)$ under low LAI conditions potentially influenced by human effects (a high water use efficiency and less constrained by environmental stresses). Please check Wei et al. (2017) for further information.

*Thank you for noting this, we added Wei et al. (2017) to this subsection in addition to its mention in other subsections given its importance to this topic.*

9. P17L25: Please also introduce the measurement of radiometric surface temperature.

*This comment made us realize that we used a subscripted 'T' when discussing temperature in the Appendix, which changed to 'Temp' to avoid confusion. We did not want to delve into too much detail for a well-established approach, especially given the diversity of platforms from which radiometric surface temperature can be measured, so we kept the wording as is.*

10. P18L19 gsurf: g_{surf}

*Thank you for noting this error. This section has since been re-written by Dr. Rigden.*

11. P16L14 (Fisher et al., 2008): Fisher et al. (2008)
12. P16L15 (Jarvis, 1976) : Jarvis (1976)
13. P17L24 Norman et al. )1995): Norman et al. (1995)

*Thank you for noting these errors that we made when referencing.*

---

## Author Comment (AC4) · 19 Jun 2019

We were fortunate in the open review process to receive important technical inputs by an expert, Dr. Anne Klosterhalfen. As a consequence, we invited Dr. Klosterhalfen to further improve the subsection on flux-variance similarity and the manuscript as a whole, and we felt that her intellectual contributions warranted coauthorship. In the original submission we included a subsection that described the work of Dr. Angela Rigden, and subsequently felt that it would be more appropriate to have Dr. Rigden write this subsection with the input of other authors rather than describe her work without her input. We followed a strategy in which experts were primarily responsible for

subsections of the manuscript and these subsections were subsequently edited by coauthors. By adding Dr. Rigden as a coauthor we were able to apply this coauthorship structure consistently throughout the manuscript. We also sought advice about novel approaches to partition evaporation and transpiration for which Dr. Miriam Coenders-Gerrits is an expert. We solicited her opinions on the future directions section and felt that her valuable advice throughout the manuscript and attention to detail dramatically improved the manuscript and warranted coauthorship.

———————————————————

---

## Author Response (AR2)

Dear Dr. De Kauwe,

Please find below our responses to Referee comments for bg-2019-85. We agree that more references to insightful modeling papers improve our manuscript, but we also want to maintain focus on the challenges of measuring ecosystem-scale E and T so that models have adequate observations for validation and critique. We added a number of references suggested by Reviewer #3 to detail the important findings of recent modeling approaches but decided that a comprehensive analysis of modeling results would distract from the present analysis. We also prepared a response to your comments on the previous version of the manuscript and realized that this may not have been uploaded into the Copernicus system. Our original letter is pasted below for completeness.

Sincerely,

Tarek El Madany

Reviewer #3
Review of Stoy et al.

This manuscript is a review of transpiration partitioning approaches. I came in late(r) in the review process and was not able to provide a full review. In my brief review I note that modeling studies are almost completely absent from the discussion. There are several modeling approaches that have worked to understand discrepancies in transpiration partitioning and the authors do not review them. Figure 1 is a start, but I find that plots like this are not very helpful (think of the original PILPS studies where plots of an OM difference were not helpful, but groupings or tables of LSM process were). Something like a replicate of the table in Schlesinger and Jasechko, 2014 is a start, but perhaps this could be updated to go beyond simply changes in canopy fraction S 2.1), which is a bit of a naive approach in my opinion and misses actual processes in modeling transpiration. I think this manuscript would be improved with an additional section discussion the advances and modeling of T/ET in a process based manner, with a table similar to Table 1. Some suggested references to get the authors started are included below.

*We to focus mostly on observational studies of transpiration and evaporation and current and emerging measurements for partitioning them. Without adequate observations, models cannot be adequately validated. That being said, the Reviewer makes an interesting point and some of the references that are noted below point to the idea that inadequate lateral flow representation in current LSMs are likely a major reason for model/measurement mismatches. This aligns with the comments of Dr. De Kauwe on a previous version of the manuscript, and we note that we may not have submitted a response letter to these comments. We have attached the original letter below.*

*Figure 1 is but one way of noting some of the important discrepancies that arise between ET models and measurements. It focuses on seasonal patterns, and the paper by Matheny et al. (2014) focuses on diurnal patterns. To our knowledge, no study has yet performed a comprehensive analysis of model performance across multiple time scales using the NACP Site-Level Interim Synthesis. Such an analysis would add unnecessary length to the present paper, but we agree that it would be interesting. We also feel that a comprehensive discussion of the treatment of T and E in models would make for a compelling stand-alone manuscript, but would distract from the present study, which includes 278 references before adding the interesting references that were suggested below. We thank the Referee for their comments and provide a brief discussion of all of the suggested references, including justification for not including them in two instances. Most references are now included in section 2.1.*

References
Shrestha et al Effects of horizontal grid resolution on evapotranspiration partitioning using TerrSysMP JoH 2017

*This study finds that larger model grid sizes result in more evaporation, suggesting that sub-grid processes including hydrologic redistribution are critical for simulating T/ET dynamics.*

Chang et al Why Do Large-Scale Land Surface Models Produce a Low Ratio of Transpiration to Evapotranspiration? JGR: Atmospheres 2018

*This interesting manuscript was published after we began work on the present manuscript and thank you for pointing it out to us. The authors find that realistic lateral flow simulation creates situations where soil evaporation is suppressed in favor of transpiration from deeper water sources. We now cite this reference as well as the Ji et al and Fang et al. studies in the modeling section that discusses the discrepancy between models and measurements.*

Maxwell and Condon Connections between groundwater flow and transpiration partitioning Science 2016

*Maxwell and Condon find that lateral flow and groundwater dynamics are critical for simulating transpiration and we now cite this important paper.*

Clark et al The evolution of process-based hydrologic models: historical challenges and the collective quest for physical realism HESS 2017

*This manuscript references evapotranspiration partitioning once in reference to the Maxwell and Condon (2016) paper that we now cite.*

Fatichi and Pappas Constrained variability of modeled T:ET ratio across biomes GRL 2017

*We cited this interesting study extensively in the manuscript.*

Rogers et al A roadmap for improving the representation of photosynthesis in Earth system models New Phytologist 2017

*This manuscript makes a number of key recommendations regarding photosynthesis modeling, one of which is to include the sensitivity of stomatal conductance to vapor pressure deficit. We added this reference to section 2.2 that deals with VPD sensitivity.*

Ji et al Do Lateral Flows Matter for the Hyperresolution Land Surface Modeling? GRL 2017

*This paper emphasizes the importance of lateral flow simulation to transpiration and evaporation partitioning, especially during dry conditions.*

Fang et al Influence of landscape heterogeneity on water available to tropical forests in an Amazonian catchment and implications for modeling drought response JGR Atmospheres 2017

*This manuscript studies drought in the Amazon and finds an important role of the wilting parameter in the ACME land model.*

Han et al Hydroclimatic response of evapotranspiration partitioning to prolonged droughts in semiarid grassland JOH 2018

*This manuscript emphasizes important differences in T/ET estimates that result in response to drought in grassland ecosystems. It follows the underlying water use efficiency (uWUE) approach that we describe in detail in section 3.1. We added it to this section.*

Referee #4

This manuscript is a comprehensive and very useful review of methods for partitioning evaporation and transpiration in terrestrial ecosystems. It is well written and organized, and generally does a good job of explaining why these techniques are important, what their strengths and weaknesses are, and how their application might be improved in the future. Overall I thought it was an excellent review and will be a very useful addition to the literature.

*Thank you for your support of the manuscript and for the insightful comments which improved it.*

I have a few minor comments:

Page 4, line 9: I think this should read "conductance related TO soil evaporation"

*This is correct, we edited the text.*

Page 10, line 18: It's not clear what specifically is unprecedented here. Are the scales of current SIF measurements unprecedented with respect to previous SIF measurements? Or is SIF unprecedented with respect to other measurement techniques in its potential for high spatial and temporal resolution measurements?

*We agree with you that 'unprecedented' is a qualitative superlative that should be avoided. We felt that 'multiple' was an accurate descriptor of a measurement of fluourescence, which can theoretically be measured at the scale of a single photon.*

Page 11, line 10: Typos in "canopy scaling" and "they aerodynamic conductance"

*We corrected the typo in 'canopy' and removed 'they'.*

Page 15, line 10-20: I would make sure it is clear that the regression is not supposed to be fitting the blue dots in Figure 4, and maybe also emphasize it in the caption to the figure. At first glance it looks like the lines are very poor fits to the cloud of dots, and it took me a minute to realize that this was the wrong interpretation of the figure. Perhaps dots below the 95% level could be plotted in a lighter color to help show that the regression is not supposed to be fitting the whole cloud?

*This is a good point and something that we struggled with a bit when creating the figure. The regression is meant to be a boundary line fit. We changed the color of the eddy covariance measurements to gray and now explicitly describe these in the figure legend. We also edited the legend for clarity.*

Page 17, line 6-7: This sentence refers to local overpass times, but never specifies what location this is for. Overpass times would be different depending on latitude, so this must be for a specific place or a particular latitude.

*This is correct; it is not possible to take a snapshot of an entire time zone at once. We removed the reference to the specific time and now just note 'morning' and 'afternoon'.*

Section 4.2: This section is based on a new application of two versions of the flux-variance partitioning approach to a flux site. The technique itself is well documented in the referenced papers in this section, but there are some analysis choices and preprocessing steps associated with applying techniques like this to a new site. It might be a good idea to include a more detailed description of how the technique was applied to the site in an appendix or supplement. Alternately, making the analysis code for this section available in a public repository would allow this section to be evaluated more thoroughly and/or replicated by interested readers.

*The latter analysis is based on the FluxPart algorithm available at [https://pypi.org/project/fluxpart/](https://pypi.org/project/fluxpart/). The innovation is the addition of a routine for closed path infrared gas analyzers, which will be released soon. In our opinion, the most critical factor when specifying how the algorithms were employed for a proof-of-concept test is the treatment of water use efficiency. Water use efficiency was estimated by the algorithm in both instances, rather than specified. We have adjusted the text accordingly.*

Comments to the Author:
Dear Authors,

the reviews of your manuscript were extremely positive, I have decided that minor revisions are necessary before the manuscript can be published. I agree with the reviewers that this manuscript is very readable and will likely be well received by the community. In revising your manuscript to address the various issues highlighted by the reviewers can I ask you to consider a few points from me as well.

*Thank you for your support of the manuscript and for your insightful comments, which we address below.*

1. The algorithm descriptions covered in the appendix are extremely useful for the community. I note that it is referred to once in the introduction but it would be great if you could find at least one more place in the manuscript to refer to the appendix (perhaps the discussion?). I think what you've done here is very valuable and it would be a shame for a reader to miss this.

*Thank you for this suggestion; we did put quite a bit of work into the Appendix but felt that it added too much length to the text. We refer to the Appendix in the Introduction, and in section 3.1 and 3.3. We added another reference in section 3 and in the Conclusions section.*

2. In 2.1 where you talk about the CMIP models having a T/ET ratio of 0.22-0.58 I think it would be valuable to offer some insight into why: (a) they disagree with each other; and (b) why this ratio is noticeably below other data-based estimates. This is an optional suggestion but I do think given this is a review it would be good to inform the reader. Perhaps they might consider citing Berg and Sheffield - Evapotranspiration Partitioning in CMIP5 Models: Uncertainties and Future Projections. Similarly, you might wish to more explicitly raise the issue of discrepancies amongst how models simulate LAI and the impact this has on the water cycle / ET partitioning.

*We would also like to know more about the reasons for the discrepancy but Wei et al. (2017) only note that the reason is due to methodological differences without discussing in detail why, perhaps due to the short format of Geophysical Research Letters. Additional explanations are also not available in the supplement of Wei et al. (2017). This strikes us as an important avenue of future research.*

**Note that we now include a brief description of modeling results in response to Referee #3.**

3. Again feel free to ignore this, but this is one of the few papers I've seen raise this issue. The authors neatly raise the issue of interception. In our 2013 GCB paper on WUE (Forest water use and water use efficiency at elevated CO2: a model-data intercomparison at two contrasting temperate forest FACE sites), we found that the proportion of intercepted water varied among the models by between 2-14%. This was considerably below the field estimates for the sites (and the range you quote in 3.6). It was striking how data free the assumptions were than underpinned how interception is treated in models. I'm not suggesting you get into how models simulate interception, I just think you might consider highlighting this is a serious problem for models and

may contribute to erroneous partitioning ratios - see above.

*Intercepted water is very difficult to measure and we were fortunate to have an expert (Dr. Shuguang Liu) help with a subsection on it. We added the findings of De Kauwe et al. 2013 to further emphasize its importance for models.*

3. The paper didn't seem to make much of soil evaporation? I realise it is a minor component of total ET, but recently we noted how poorly this was simulated by models. In a water-limited, semi-arid ecosystem, some models thought soil evaporation was around 50-130 mm yr-1, whilst other models thought it was 2-3.5 times greater (Challenging terrestrial biosphere models with data from the long-term multifactor Prairie Heating and CO2 Enrichment experiment). I only raise this example because it suggests to me that this isn't a trivial process to model (otherwise there wouldn't be this disagreement). I was expecting to see some sub-section on soil evaporation, but this may simply be my personal bias on this issue, so ignore as you wish.

*Following referee comments, we now added a section on soil evaporation following the new manuscript by Or and Lehman (2019). The reason for our very brief discussion of soil evaporation before is that other reviews have covered it. Now, with new analytical techniques for estimating it, we agree with the reviewer and added a subsection to the manuscript and cite De Kauwe et al. 2017.*

4. In table 2 where the variability in the exponential term is shown across models, I feel whilst interesting - without some context or explanation, it is a bit limited in terms of insight. Could the authors group the models by their stomatal assumptions (Ball-Berry, Leuning, etc). Does this help explain why they vary? Why is the BEPS model most similar to the observations?

*We are not entirely sure why BEPS is most similar to observations but find that it is interesting that it does. We did not want to pursue a long intercomparison of models versus measurements in the present manuscript, and this comment combined with the comment above made us realize that a stand-alone multi-model intercomparison of CMIP5 and other models with respect to evaporation and transpiration partitioning would be forthcoming.*

5. Was there a reason the ECOSTRESS mission wasn't mentioned in section 5?

*We now mention ECOSTRESS explicitly in section 5.*

6. When the authors discuss partitioning via the use of GPP in the WUE approaches, it would be worth mentioning that GPP isn't strictly an observation (so any errors in GPP will propagate here).

*Thank you for pointing this out, we added a passage about GPP uncertainty to section 3.*

Best wishes,

Martin